# Scalable Gaussian Processes with Latent Kronecker Structure

**Jihao Andreas Lin** [1 2 3]  **Sebastian Ament** [1]  **Maximilian Balandat** [1]  **David Eriksson** [1]
**José Miguel Hernández-Lobato** [2]  **Eytan Bakshy** [1]

## Abstract

Applying Gaussian processes (GPs) to very large datasets remains a challenge due to limited computational scalability. Matrix structures, such as the Kronecker product, can accelerate operations significantly, but their application commonly entails approximations or unrealistic assumptions. In particular, the most common path to creating a Kronecker-structured kernel matrix is by evaluating a product kernel on gridded inputs that can be expressed as a Cartesian product. However, this structure is lost if any observation is missing, breaking the Cartesian product structure, which frequently occurs in real-world data such as time series. To address this limitation, we propose leveraging *latent Kronecker structure*, by expressing the kernel matrix of observed values as the projection of a latent Kronecker product. In combination with iterative linear system solvers and pathwise conditioning, our method facilitates inference of exact GPs while requiring substantially fewer computational resources than standard iterative methods. We demonstrate that our method outperforms state-of-the-art sparse and variational GPs on real-world datasets with up to five million examples, including robotics, automated machine learning, and climate applications.

## 1. Introduction

Gaussian processes (GPs) are probabilistic models prized for their flexibility, data efficiency, and well-calibrated uncertainty estimates. They play a key role in many applications such as Bayesian optimization (Frazier, 2018; Garnett, 2023), reinforcement learning (Deisenroth & Rasmussen, 2011), and active learning (Riis et al., 2022). However, exact

---

[1]Meta [2]University of Cambridge [3]Max Planck Institute for Intelligent Systems, Tübingen. Correspondence to: Jihao Andreas Lin <jandylin@meta.com>.

*Proceedings of the 42$^{nd}$ International Conference on Machine Learning*, Vancouver, Canada. PMLR 267, 2025. Copyright 2025 by the author(s).

GPs are notoriously challenging to scale to large numbers of training examples $n$. This primarily stems from having to solve an $n \times n$ linear system involving the kernel matrix to compute the marginal likelihood and the posterior, which has $\mathcal{O}(n^3)$ time complexity using direct methods.

To address these scalability challenges, a plethora of approaches have been proposed, which usually fall into two main categories. *Sparse* GP approaches reduce the size of the underlying linear system by introducing a set of *inducing points* to approximate the full GP; they include conventional (Quinonero-Candela & Rasmussen, 2005) and variational (Titsias, 2009) formulations. *Iterative* approaches (Gardner et al., 2018a; Wang et al., 2019) employ solvers such as the linear conjugate gradient method, often leveraging hardware parallelism and structured kernel matrices.

In particular, Kronecker product structure permits efficient matrix-vector multiplication (MVM). This structure arises when a product kernel is evaluated on data points from a Cartesian product space. A common (though not the only) example of this is spatiotemporal data, as found in climate science for instance, where a quantity of interest, such as temperature or humidity, varies across space and is measured at regular time intervals. In particular, assuming observations collected at $p$ locations and $q$ times result in $pq$ data points in total, the kernel matrix generated by evaluating a product kernel on this data can then be expressed as the Kronecker product of two smaller $p \times p$ and $q \times q$ matrices. This allows MVM in $\mathcal{O}(p^2q + pq^2)$ instead of $\mathcal{O}(p^2q^2)$ time and $\mathcal{O}(p^2 + q^2)$ instead of $\mathcal{O}(p^2q^2)$ space, such that iterative methods require substantially less time and memory. However, in real-world data, this structure frequently breaks due to missing observations, typically requiring a reversion to generic sparse or iterative methods that either do not take advantage of special structure or entail approximations to the matrix.

In this paper, we propose *latent Kronecker structure*, which enables efficient inference despite missing values by representing the joint covariance matrix of observed values as a lazily-evaluated product between projection matrices and a latent Kronecker product. Combined with iterative methods, which otherwise would not be able to leverage Kronecker structure in the presence of missing values, this reduces the asymptotic time complexity from $\mathcal{O}(p^2q^2)$ to $\mathcal{O}(p^2q + pq^2)$,

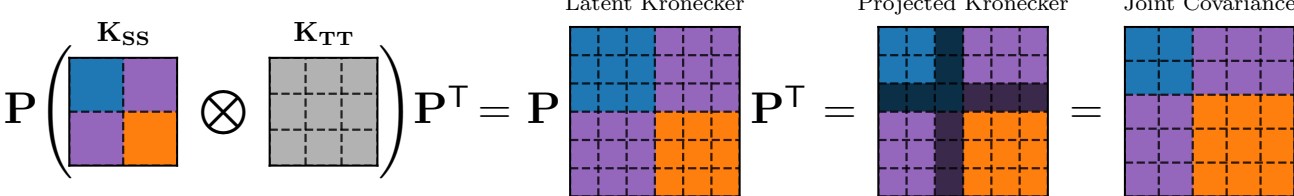

*Figure 1.* Illustrative example considering input data points $\{(\mathbf{s}_1, \mathbf{t}_1), (\mathbf{s}_1, \mathbf{t}_2), (\mathbf{s}_2, \mathbf{t}_1), (\mathbf{s}_2, \mathbf{t}_2), (\mathbf{s}_2, \mathbf{t}_3)\}$, consisting of two out of three time steps at spatial location $\mathbf{s}_1$ and three out of three time steps at spatial location $\mathbf{s}_2$. Due to the missing observation at $(\mathbf{s}_1, \mathbf{t}_3)$, the inputs cannot be expressed as a Cartesian product and the joint covariance matrix cannot be directly expressed as a Kronecker product. *Latent* Kronecker structure assumes that missing observations exist, such that a *latent* Kronecker product can be leveraged for computational benefits. The original joint covariance matrix is exactly recovered by removing the missing rows and columns via projections.

and space complexity from $\mathcal{O}(p^2 q^2)$ to $\mathcal{O}(p^2 + q^2)$. Importantly, unlike sparse approaches, our method does not introduce approximations of the GP prior. This avoids common problems of sparse GPs, such as underestimating predictive variances (Jankowiak et al., 2020), and overestimating noise (Titsias, 2009; Bauer et al., 2016) due to limited model expressivity with a constant number of inducing points.

In contrast, our method, Latent Kronecker GP (LKGP), facilitates highly scalable inference of *exact* GP models with product kernels. We empirically demonstrate the superior computational scalability, speed, and modeling performance of our approach on various real-world applications including inverse dynamics prediction for robotics, learning curve prediction, and climate modeling, using datasets with up to five million data points.

## 2. Background and Related Work

Let $f : \mathcal{X} \to \mathbb{R}$ be a stochastic process such that for any finite subset $\{\mathbf{x}_i\}_{i=1}^n \subset \mathcal{X}$, the collection of random variables $\{f(\mathbf{x}_i)\}_{i=1}^n$ follows a multivariate normal distribution. This property defines $f$ as a Gaussian process (GP), which is uniquely identified by its mean function $\mu(\cdot) = \mathbb{E}[f(\cdot)]$ and its kernel function $k(\cdot, \cdot') = \mathrm{Cov}(f(\cdot), f(\cdot'))$. We suppress the dependence on kernel hyperparameters for brevity.

To perform GP regression, let the training data consist of inputs $\{\mathbf{x}_i\}_{i=1}^n = \mathbf{X} \subset \mathcal{X}$ and outputs $\{y_i\}_{i=1}^n = \mathbf{y} \in \mathbb{R}^n$. We consider the Bayesian model $y_i = f(\mathbf{x}_i) + \epsilon_i$, where $f \sim \mathrm{GP}(\mu, k)$ and each $\epsilon_i \sim \mathcal{N}(0, \sigma^2)$. Without loss of generality, we assume $\mu \equiv 0$. The posterior of this model is given by $f|\mathbf{y} \sim \mathrm{GP}(\mu_{f|\mathbf{y}}, k_{f|\mathbf{y}})$, with

$$\begin{aligned} \mu_{f|\mathbf{y}}(\cdot) &= \mathbf{K}_{(\cdot)\mathbf{X}}(\mathbf{K}_{\mathbf{X}\mathbf{X}} + \sigma^2 \mathbf{I})^{-1}\mathbf{y}, \\ k_{f|\mathbf{y}}(\cdot, \cdot') &= \mathbf{K}_{(\cdot, \cdot')} - \mathbf{K}_{(\cdot)\mathbf{X}}(\mathbf{K}_{\mathbf{X}\mathbf{X}} + \sigma^2 \mathbf{I})^{-1}\mathbf{K}_{\mathbf{X}(\cdot')}, \end{aligned} \quad (1)$$

where $\mathbf{K}_{\mathbf{X}\mathbf{X}}$ denotes the covariance matrix $[k(\mathbf{x}_i, \mathbf{x}_j)]_{i,j=1}^n$ and $\mathbf{K}_{(\cdot)\mathbf{X}}$, $\mathbf{K}_{\mathbf{X}(\cdot')}$, and $\mathbf{K}_{(\cdot, \cdot')}$ denote pairwise evaluations of the same kind. For a comprehensive introduction to GPs, we refer to Rasmussen & Williams (2006).

**Pathwise Conditioning**  Instead of computing the mean and covariance of the posterior distribution (1), Wilson et al. (2020; 2021) directly express a sample from the GP posterior as a random function

$$(f|\mathbf{y})(\cdot) = f(\cdot) + \mathbf{K}_{(\cdot)\mathbf{X}}(\mathbf{K}_{\mathbf{X}\mathbf{X}} + \sigma^2 \mathbf{I})^{-1}(\mathbf{y} - (\mathbf{f}_{\mathbf{X}} + \boldsymbol{\epsilon})),$$

where $f \sim \mathrm{GP}(\mu, k)$ is a sample from the GP prior, $\mathbf{f}_{\mathbf{X}}$ is its evaluation at the training data $\mathbf{X}$, and $\boldsymbol{\epsilon} \sim \mathcal{N}(\mathbf{0}, \sigma^2 \mathbf{I})$ is a random vector. For product kernels on Cartesian product spaces, pathwise conditioning improves the asymptotic complexity of drawing samples from the GP posterior (Maddox et al., 2021). We leverage this property in Section 3.

**Sparse and Variational Methods**  Conventional GP implementations scale poorly with the number of training examples $n$ because posterior inference (1) requires a linear solve against the kernel matrix, which scales as $\mathcal{O}(n^3)$ using direct methods. To address this, a large number of *sparse* GP (Snelson & Ghahramani, 2005) approaches that rely on approximating the kernel matrix at a set of *inducing points*, have been proposed. These methods typically reduce the training cost from $\mathcal{O}(n^3)$ to $\mathcal{O}(n^2 m + m^3)$ per gradient step where $m$ is the number of inducing points and $m \ll n$.

Sparse Gaussian Process Regression (SGPR) is an inducing point method which introduces variational inference for sparse GPs (Titsias, 2009). The locations of the inducing points are optimized along with the kernel hyperparameters using the evidence lower bound (ELBO). Both the ELBO optimization and posterior inference require $\mathcal{O}(nm^2)$, which is significantly faster than exact GP inference when $m \ll n$.

Stochastic Variational Gaussian Processes (SVGP) improve the computational complexity of SGPR by decomposing the ELBO into a sum of losses over the training labels (Hensman et al., 2013). This makes it possible to leverage stochastic gradient-based optimization for training which reduces the computational complexity to $\mathcal{O}(m^3)$. Consequently, SVGP can be scaled to very large datasets as the asymptotic training complexity no longer depends on $n$.

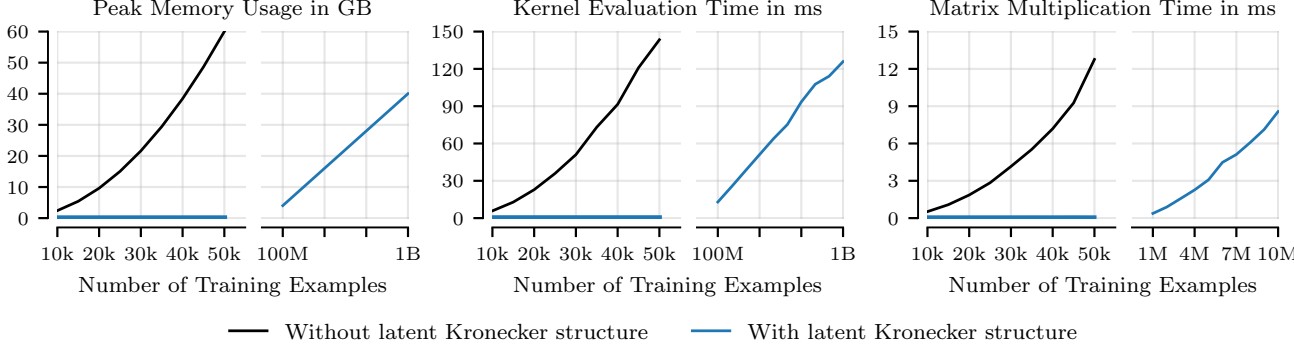

*Figure 2.* Illustration of computational resources used during kernel evaluation and matrix multiplication on ten-dimensional synthetic datasets of different sizes. Without latent Kronecker structure, memory usage escalates quickly and kernel evaluation time dominates matrix multiplication time asymptotically. With latent Kronecker structure, computations can be scaled to several orders of magnitude larger datasets under similar computational resource usage, and matrix multiplication time dominates kernel evaluation time asymptotically. Reported results are the mean over one hundred repetitions using a squared exponential kernel.

Wu et al. (2022) proposed the Variational Nearest Neighbor Gaussian Process (VNNGP), which introduces a prior that only retains correlations for $K$ nearest neighbors. This results in a sparse kernel matrix where the time complexity of estimating the ELBO is $\mathcal{O}((n_b + m_b)K^3)$ for a mini-batch of size $n_b$ and a mini-batch of $m_b$ inducing points.

Wenger et al. (2024) introduced Computation-Aware Gaussian Processes (CaGP), a class of GP approximation methods which leverage low-rank approximations while dealing with the overconfidence problems of SVGP, guaranteeing that the posterior variance of CaGP is always greater than the posterior variance of an exact GP. The additional variance is interpreted as *computational uncertainty* and helps the method achieve well-calibrated predictions while scaling to large datasets.

While the above methods scale to large datasets, they are limited by performing inference of approximate GP models.

**Iterative Linear System Solvers**  Alternatively, costly, or even intractable, direct solves of large linear systems can be avoided by leveraging iterative linear system solvers. To this end, the solution $\mathbf{v} = \mathbf{A}^{-1}\mathbf{b}$ to the linear system $\mathbf{A}\mathbf{v} = \mathbf{b}$ with positive-definite coefficient matrix $\mathbf{A}$ is instead equivalently expressed as the unique optimum of a convex quadratic objective,

$$\mathbf{v} = \mathbf{A}^{-1}\mathbf{b} \iff \mathbf{v} = \arg\min_{\mathbf{u}} \quad \frac{1}{2}\mathbf{u}^{\mathsf{T}}\mathbf{A}\mathbf{u} - \mathbf{u}^{\mathsf{T}}\mathbf{b},$$

which facilitates the use of iterative optimization algorithms to solve large linear systems in a scalable way using MVMs. In the context of GPs, setting $\mathbf{A}$ to $\mathbf{K}_{\mathbf{XX}} + \sigma^2\mathbf{I}$ and $\mathbf{b}$ to $\mathbf{y}$ produces the posterior mean $\mu_{f|\mathbf{y}}(\cdot)$ as $\mathbf{K}_{(\cdot)\mathbf{X}}\mathbf{v}$. Setting $\mathbf{b}$ to $\mathbf{y} - (\mathbf{f}_{\mathbf{X}} + \boldsymbol{\epsilon})$ generates samples from the GP posterior via pathwise conditioning as $f(\cdot) + \mathbf{K}_{(\cdot)\mathbf{X}}\mathbf{v}$. Other choices

of $\mathbf{b}$ enable the estimation of marginal likelihood gradients for kernel hyperparameter optimization (Lin et al., 2024b). In terms of actual optimization algorithms used as iterative linear system solvers for GPs, Gardner et al. (2018a) popularized conjugate gradients, Wu et al. (2024) introduced alternating projections, and Lin et al. (2023; 2024a) applied stochastic gradient descent.

In contrast to sparse and variational approaches, iterative methods perform inference of the exact GP model up to a certain numerical tolerance. We leverage iterative linear system solvers for efficient inference in Section 3.

**Structured Kernels**  In the special case of a product kernels with multidimensional data on a Cartesian grid, the kernel matrix decomposes into a Kronecker product of one-dimensional kernels: $\mathbf{K} = \mathbf{K_1} \otimes \ldots \otimes \mathbf{K_d}$. This allows expressing matrix decompositions of $\mathbf{K}$ (including Cholesky and eigendecompositions) as individual decompositions of $\mathbf{K_1}, \ldots, \mathbf{K_d}$, enabling efficient exact GP inference (Bonilla et al., 2007; Stegle et al., 2011; Golub & Van Loan, 2013), including for gradient observations (Ament & Gomes, 2022). To complete a partially observed grid of observations, Saatçi (2012) and Wilson et al. (2014) propose adding *imaginary observations* with a large noise variance, giving rise to an approximation which only converges as the artificial noise variance goes to infinity and leads to ill-conditioning.

**Structured Kernel Approximations**  In the case of non-gridded inputs, we can leverage the Structured Kernel Interpolation (SKI) kernel to approximate the original kernel: $\mathbf{K_{XX}} \approx \mathbf{W}\mathbf{K_{UU}}\mathbf{W}^{\mathsf{T}}$, where $\mathbf{W}$ is a matrix of interpolation weights and $\mathbf{K_{UU}}$ exhibits structure such as Toeplitz or Kronecker (Wilson & Nickisch, 2015). This algebraic structure can be used for efficient kernel learning using

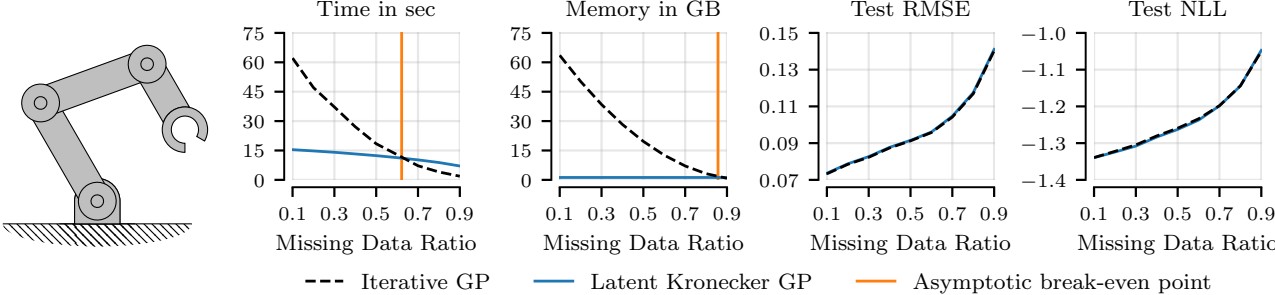

*Figure 3.* Predicting the inverse dynamics of an anthropomorphic robot arm with seven degrees of freedom. Compared to standard iterative methods, leveraging latent Kronecker structure results in significantly lower runtime and memory requirements while maintaining the same predictive performance. The asymptotic break-even point, at which both methods asymptotically require the same amount of time or memory, closely matches the empirical break-even point. Reported results are the mean over 10 different random splits of the data.

conjugate gradients for linear solves and Lanczos to approximate the log determinant (Dong et al., 2017). SKI is limited to low-dimensional inputs as the number of inducing points $m$ increases exponentially with the input dimensionality.

To address the scalability issues of SKI with respect to the number of input dimensions, Gardner et al. (2018b) proposed Structured Kernel Interpolation for Products (SKIP), which approximates the original kernel by a Hadamard product of one-dimensional SKI kernels. Every SKI kernel is further approximated by a low-rank Lanczos decomposition, which gives rise to an algebraic structure that allows for efficient MVMs: $\mathbf{K_{XX}} \approx (\mathbf{Q_1 T_1 Q_1}^\mathsf{T}) \odot \ldots \odot (\mathbf{Q_d T_d Q_d}^\mathsf{T})$. While SKIP addresses the scalability issues of SKI, it is usually not competitive with other methods due to the errors introduced by the approximations (Kapoor et al., 2021).

Hierarchical matrix approximation algorithms constitute another notable class of methods. Ambikasaran et al. (2016) introduced an approach which is based on purely algebraic approximations and permits direct matrix inversion. Ryan et al. (2022) proposed an algorithm which uses automatic differentiation and automated symbolic computations which leverage the analytical structure of the underlying kernel. The main limitation of this class of methods is also scalability to higher-dimensional spaces, for which the hierarchical approximations become inefficient.

**State-Space Approaches** Hartikainen & Särkkä (2010); Sarkka et al. (2013) introduced state-space formulations for (spatio-)temporal GP regression, whose computation has a linear complexity with respect to the number of temporal observations for a given point in space. See Särkkä & Svensson (2023) for a review of the methods. Recently, Laplante et al. (2025) leveraged this formulation to accelerate robust spatiotemporal GP regression. Pförtner et al. (2025) extended the computation-aware framework of Wenger et al. (2024) to quantify errors caused by approximate filtering and smoothing methods. The main limitations of the state-

space approaches to GP regression are 1) the requirement that the temporal kernel be stationary, and 2) that they scale quadratically or cubically with the spatial dimensionality.

The method proposed herein scales linearly in the spatial dimensionality, does not require the temporal kernel to be stationary, and is not limited to spatiotemporal data. If the method is applied to spatiotemporal data collected in uniform temporal intervals, and the temporal kernel is stationary, the method can be accelerated to be quasi-linear in the number of time steps by leveraging the Toeplitz structure of the temporal kernel matrix (Wilson & Nickisch, 2015).

## 3. Latent Kronecker Structure

In this paper, we consider a Gaussian process $f : \mathcal{X} \to \mathbb{R}$ defined on a Cartesian product space $\mathcal{X} = \mathcal{S} \times \mathcal{T}$. For illustrative purposes, $\mathcal{S}$ could be associated with spatial dimensions and $\mathcal{T}$ could refer to a time dimension or a *task index*, as commonly used in multi-task GPs (Bonilla et al., 2007), but we emphasize that neither of them are limited to one-dimensional spaces. A natural way to model this problem is to define a kernel $k_\mathcal{X}$ on the product space $\mathcal{X}$, which results in a joint covariance

$$\mathrm{Cov}(f(\mathbf{x}), f(\mathbf{x}')) = k_\mathcal{X}(\mathbf{x}, \mathbf{x}') = k_\mathcal{X}((\mathbf{s}, \mathbf{t}), (\mathbf{s}', \mathbf{t}')),$$

where $\mathbf{s}, \mathbf{s}' \in \mathcal{S}$ and $\mathbf{t}, \mathbf{t}' \in \mathcal{T}$. However, this results in scalability issues when performing GP regression. If the training data consists of $n$ outputs $\{y_i\}_{i=1}^n = \mathbf{y} \in \mathbb{R}^n$ observed at $p$ spatial locations $\{\mathbf{s}_j\}_{j=1}^p = \mathbf{S} \subset \mathcal{S}$ and $q$ time steps or tasks $\{\mathbf{t}_k\}_{k=1}^q = \mathbf{T} \subset \mathcal{T}$, such that $n = pq$, then the joint covariance matrix requires $\mathcal{O}(p^2 q^2)$ space and computing its Cholesky factor takes $\mathcal{O}(p^3 q^3)$ time.

**Ordinary Kronecker Structure** A common way to reduce the computational complexity in such settings is to introduce product kernels and Kronecker structure (Bonilla

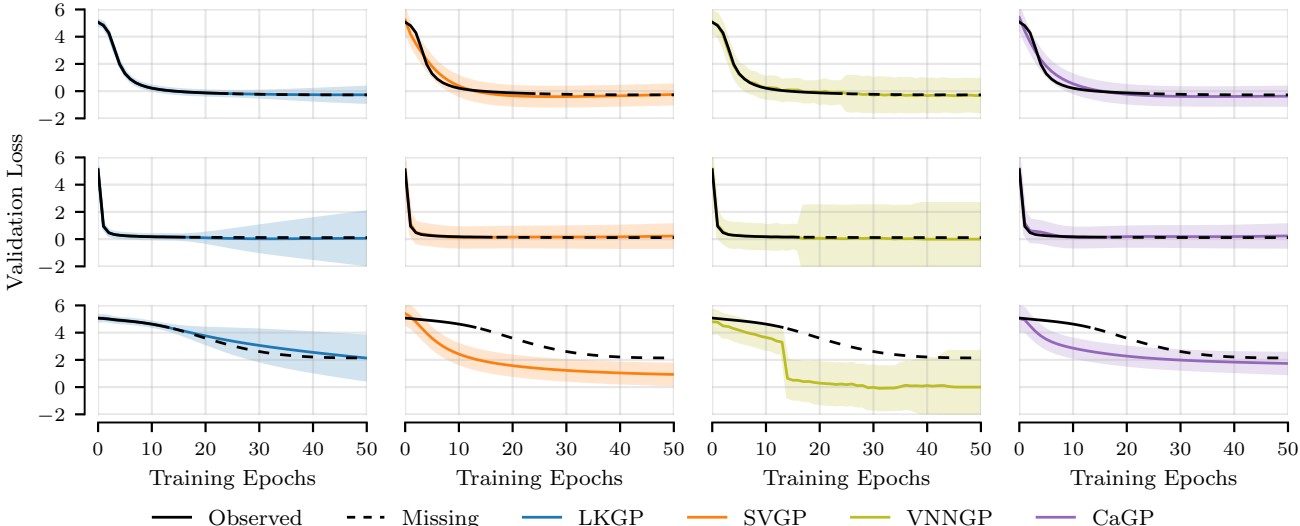

*Figure 4.* Learning curve prediction on the Fashion-MNIST data from the LCBench benchmark (Zimmer et al., 2021). Partially observed learning curves are extrapolated into the future. The predictive mean and two standard deviations of various GP models are visualized. All models are reasonably accurate in the mean, but LKGP produces the most sensible uncertainty estimates, starting with low uncertainty in the observed part of the learning curve and gradually increasing the predicted uncertainty into the missing part. The third row illustrates an outlier with significantly different behavior than most other learning curves. The sparse methods struggle because the limited number of inducing points is unlikely to cover such a case. LKGP can adapt well because it performs inference in the exact GP model.

et al., 2007; Stegle et al., 2011; Zhe et al., 2019). Defining

$$k_{\mathcal{X}}(\mathbf{x}, \mathbf{x}') = k_{\mathcal{X}}((\mathbf{s}, \mathbf{t}), (\mathbf{s}', \mathbf{t}')) = k_{\mathcal{S}}(\mathbf{s}, \mathbf{s}') \, k_{\mathcal{T}}(\mathbf{t}, \mathbf{t}'),$$

where $k_{\mathcal{S}}$ only considers spatial locations $\mathbf{s}_j$ and $k_{\mathcal{T}}$ solely acts on time steps or tasks $\mathbf{t}_k$, allowing the joint covariance matrix $\mathbf{K_{XX}}$ to factorize as

$$\underset{n \times n}{\mathbf{K_{XX}}} = k_{\mathcal{S}}(\mathbf{S}, \mathbf{S}) \otimes k_{\mathcal{T}}(\mathbf{T}, \mathbf{T}) = \underset{p \times p}{\mathbf{K_{SS}}} \otimes \underset{q \times q}{\mathbf{K_{TT}}},$$

which can be exploited by expressing decompositions of $\mathbf{K_{XX}}$ in terms of decompositions of $\mathbf{K_{SS}}$ and $\mathbf{K_{TT}}$ instead. This reduces the asymptotic time complexity to $\mathcal{O}(p^3 + q^3)$ and space complexity to $\mathcal{O}(p^2 + q^2)$.

However, the joint covariance matrix $\mathbf{K_{XX}}$ only exhibits the Kronecker structure if observations $y_i$ are available for each time step or task $\mathbf{t}_k$ at every spatial location $\mathbf{s}_j$, and this is often not the case. For example, suppose we are considering temperatures $y_i$ on different days $\mathbf{t}_k$ measured by various weather stations at locations $\mathbf{s}_j$. If a single weather station does not record the temperature on any given day, then the resulting observations are no longer fully gridded, and thus ordinary Kronecker methods cannot be applied. These *missing values* refer to missing output observations which correspond to inputs from a Cartesian product space $\mathcal{X} = \mathcal{S} \times \mathcal{T}$. They do not refer to *missing features* in the input data, and imputing missing values is not the main goal of our contribution. We focus on improving scalability by

proposing a method which facilitates the use of Kronecker products in the presence of missing values.

**Dealing with Missing Values**  In the context of gridded data with missing values, such that $n < pq$, a key insight is that the joint covariance matrix over observed values is a submatrix of the joint covariance matrix over the whole Cartesian product grid. Although the former generally does not have Kronecker structure, the latter does. We leverage this *latent* Kronecker structure in the latter matrix by expressing the joint covariance matrix over observed values as

$$\underset{n \times n}{\mathbf{K_{XX}}} = \underset{n \times pq}{\mathbf{P}} \left( \underset{p \times p}{\mathbf{K_{SS}}} \otimes \underset{q \times q}{\mathbf{K_{TT}}} \right) \underset{pq \times n}{\mathbf{P^{\mathsf{T}}}},$$

where the projection matrix $\mathbf{P}$ is constructed from the identity matrix by removing rows which correspond to missing values. See Figure 1 for an illustration. Crucially, this is not an approximation and facilitates inference of the exact GP model. In practice, we implement projections efficiently without explicitly instantiating or multiplying by $\mathbf{P}$.

**Efficient Inference via Iterative Methods**  As a result of introducing projections, the eigenvalues and eigenvectors of $\mathbf{K_{XX}}$ cannot be expressed in terms of eigenvalues and eigenvectors of $\mathbf{K_{SS}}$ and $\mathbf{K_{TT}}$ anymore, which prevents the use of Kronecker structure for efficient matrix factorization. However, despite the projections, the Kronecker structure

can still be leveraged for fast matrix multiplication. To this end, we augment the standard Kronecker product equation, $(\mathbf{A} \otimes \mathbf{B}) \operatorname{vec}(\mathbf{C}) = \operatorname{vec}(\mathbf{B}\mathbf{C}\mathbf{A}^\mathsf{T})$, with the aforementioned projections, yielding

$$\mathbf{P}(\mathbf{A} \otimes \mathbf{B})\mathbf{P}^\mathsf{T}\operatorname{vec}(\mathbf{C}) = \mathbf{P}\operatorname{vec}(\mathbf{B}\operatorname{vec}^{-1}(\mathbf{P}^\mathsf{T}\operatorname{vec}(\mathbf{C}))\mathbf{A}^\mathsf{T}).$$

In practice, vec and $\operatorname{vec}^{-1}$ are implemented as reshaping operations, $\mathbf{P}^\mathsf{T}\operatorname{vec}(\mathbf{C})$ amounts to zero padding, and left-multiplying by $\mathbf{P}$ is equivalent to slice indexing, all of which typically do not incur significant computational overheads. This facilitates efficient inference in the exact GP model via iterative methods, which only rely on MVM to compute solutions of linear systems (Gardner et al., 2018a; Wang et al., 2019). Leveraging the latent Kronecker structure reduces the asymptotic time complexity of MVM from $\mathcal{O}(n^2)$ to $\mathcal{O}(p^2q + pq^2)$, and the asymptotic space complexity from $\mathcal{O}(n^2)$ to $\mathcal{O}(p^2 + q^2)$. Furthermore, if the kernel matrix is not materialized and kernel values are instead calculated on demand, also known as lazy kernel evaluation, the asymptotic space complexity is reduced from $\mathcal{O}(n)$ to $\mathcal{O}(p + q)$.

**Posterior Samples via Pathwise Conditioning**   In the context of product kernels and gridded data, Maddox et al. (2021) proposed to draw posterior samples via pathwise conditioning (Wilson et al., 2020; 2021) to exploit Kronecker structure, reducing the asymptotic time complexity from $\mathcal{O}(p^3q^3)$ to $\mathcal{O}(p^3 + q^3)$. In our product space notation, the pathwise conditioning equation can be written as

$$(f|\mathbf{y})((\cdot_\mathbf{s}, \cdot_\mathbf{t})) = f((\cdot_\mathbf{s}, \cdot_\mathbf{t})) + (\mathbf{K}_{(\cdot_\mathbf{s})\mathbf{S}} \otimes \mathbf{K}_{(\cdot_\mathbf{t})\mathbf{T}})\,\mathbf{v},$$

where $\mathbf{v}$ is the inverse matrix-vector product

$$\mathbf{v} = (\mathbf{K}_{\mathbf{SS}} \otimes \mathbf{K}_{\mathbf{TT}} + \sigma^2\mathbf{I})^{-1}(\mathbf{y} - (\mathbf{f}_{\mathbf{S} \times \mathbf{T}} + \boldsymbol{\epsilon})),$$

and $\boldsymbol{\epsilon} \sim \mathcal{N}(\mathbf{0}, \sigma^2\mathbf{I})$. To support latent Kronecker structure, we again introduce projections such that

$$\mathbf{v} = \mathbf{P}^\mathsf{T}(\mathbf{P}(\mathbf{K}_{\mathbf{SS}} \otimes \mathbf{K}_{\mathbf{TT}})\mathbf{P}^\mathsf{T} + \sigma^2\mathbf{I})^{-1}(\mathbf{y} - (\mathbf{P}\mathbf{f}_{\mathbf{S} \times \mathbf{T}} + \boldsymbol{\epsilon})),$$

resulting in exact samples from the exact GP posterior. In combination with iterative linear system solvers, latent Kronecker structure can be exploited to compute $\mathbf{v}$ using efficient MVM. Further, computations can be cached or amortized to accelerate Bayesian optimization or marginal likelihood optimization (Lin et al., 2024b).

**Discussion of Computational Benefits**   The most apparent benefits of using latent Kronecker structure are the improved asymptotic time and space complexities of matrix multiplication, as discussed earlier. However, there are further, more subtle benefits. In particular, the number of kernel evaluations changes from $\mathcal{O}(n^2)$ evaluations of $k_\mathcal{X}$ to $\mathcal{O}(p^2)$ evaluations of $k_\mathcal{S}$ plus $\mathcal{O}(q^2)$ evaluations of $k_\mathcal{T}$. This may seem irrelevant if kernel matrices fit into memory, because,

in this case, the cost of kernel evaluations is amortized, and the total runtime is dominated by matrix multiplication. But, if kernel matrices do not fit into memory, their values must be rematerialized during each matrix multiplication, leading to many repeated kernel evaluations which may dominate the total runtime. Due to much lower memory requirements, latent Kronecker structure also raises the threshold after which kernel values must be rematerialized. Figure 2 illustrates memory usage, kernel evaluation times, and MVM times on ten-dimensional synthetic datasets of various sizes, assuming a balanced factorization $p = q = \sqrt{n}$.

**Efficiency of Latent Kronecker Structure**   Since latent Kronecker structure can be understood as padding $n \leq pq$ observations to complete a grid of $pq$ values, a key question is at which point the number of missing values $pq - n$ is large enough such that this padding becomes inefficient and dense matrices without this structure are preferable.

**Proposition 3.1.** *Let $\gamma = 1 - n/pq$ be the missing ratio, that is, the relative amount of padding required to complete a grid with $pq$ values. Let $\gamma^*$ be the asymptotic break-even point, that is, the particular missing ratio at which a kernel matrix without factorization has the same asymptotic performance as a kernel matrix with latent Kronecker structure. The asymptotic break-even points for MVM time and memory usage are, respectively,*

$$\gamma^*_{\text{time}} = 1 - \sqrt{\frac{1}{p} + \frac{1}{q}} \quad \text{and} \quad \gamma^*_{\text{mem}} = 1 - \sqrt{\frac{1}{p^2} + \frac{1}{q^2}}.$$

See Appendix A for the derivation. In Section 4, we validate these asymptotic break-even points empirically, and observe that they closely match the empirical break-even points obtained from experiments on real-world data (see Figure 3).

**Relationship to High-order GPs**   Zhe et al. (2019) proposed High-order GPs (HOGPs), which model tensor-structured outputs through a kernel defined over low-dimensional latent embeddings of each coordinate, allowing efficient computation via Kronecker algebra. HOGPs and LKGPs both leverage Kronecker products for scalability but differ fundamentally: HOGPs require a complete grid of observations and model dependencies via inferred *latent variables*, while LKGPs handle missing values via *latent projections* of a full structured kernel matrix (without inferring any latent variables).

## 4. Experiments

We conduct three distinct experiments to empirically evaluate LKGPs: inverse dynamics prediction for robotics, learning curve prediction in an AutoML setting, and prediction of missing values in spatiotemporal climate data. In the first experiment, we compare LKGP against iterative methods

Table 1. [Learning Curve Prediction] Predictive performances and total runtimes of learning curve prediction on every fifth LCBench dataset. On average, LKGP produces the best negative log-likelihood while also requiring the least time. Reported numbers are the mean over 10 random seeds. Results of the best model are boldfaced and results of the second best model are underlined. See Tables 3 to 7 in Appendix B for the results on all datasets.

| | Model | APSFailure | MiniBooNE | blood | covertype | higgs | kr-vs-kp | segment | Average Rank |
|---|---|---|---|---|---|---|---|---|---|
| Train RMSE | LKGP | **1.705 ± 0.047** | **0.047 ± 0.001** | **0.278 ± 0.017** | **0.061 ± 0.001** | **0.039 ± 0.001** | **0.008 ± 0.000** | **0.006 ± 0.000** | **1.171 ± 0.560** |
| | SVGP | 3.893 ± 0.149 | 0.201 ± 0.004 | 0.463 ± 0.009 | 0.240 ± 0.003 | 0.200 ± 0.003 | 0.101 ± 0.001 | 0.086 ± 0.001 | 2.771 ± 0.680 |
| | VNNGP | 4.920 ± 0.194 | 0.113 ± 0.001 | 0.284 ± 0.029 | 0.158 ± 0.003 | 0.130 ± 0.002 | 0.067 ± 0.001 | 0.072 ± 0.001 | 2.314 ± 0.708 |
| | CaGP | 3.972 ± 0.147 | 0.219 ± 0.005 | 0.511 ± 0.011 | 0.241 ± 0.004 | 0.208 ± 0.002 | 0.105 ± 0.002 | 0.091 ± 0.001 | 3.743 ± 0.553 |
| Test RMSE | LKGP | 2.935 ± 0.150 | 0.335 ± 0.006 | 0.747 ± 0.016 | 0.351 ± 0.008 | 0.394 ± 0.027 | 0.261 ± 0.005 | 0.152 ± 0.003 | 2.600 ± 0.901 |
| | SVGP | **2.716 ± 0.145** | **0.316 ± 0.005** | **0.540 ± 0.013** | 0.294 ± 0.005 | **0.285 ± 0.005** | **0.232 ± 0.003** | **0.145 ± 0.002** | **1.657 ± 1.068** |
| | VNNGP | 3.053 ± 0.159 | 0.705 ± 0.013 | 0.915 ± 0.017 | 0.677 ± 0.011 | 0.642 ± 0.009 | 0.591 ± 0.006 | 0.568 ± 0.005 | 3.743 ± 0.731 |
| | CaGP | 2.810 ± 0.154 | 0.325 ± 0.006 | 0.593 ± 0.015 | **0.281 ± 0.004** | 0.296 ± 0.004 | 0.239 ± 0.004 | 0.153 ± 0.002 | 2.000 ± 0.000 |
| Train NLL | LKGP | **1.955 ± 0.020** | **-1.573 ± 0.022** | **0.160 ± 0.055** | **-1.311 ± 0.017** | **-1.718 ± 0.018** | **-2.933 ± 0.005** | **-2.943 ± 0.005** | **1.029 ± 0.167** |
| | SVGP | 3.098 ± 0.100 | -0.132 ± 0.020 | 0.652 ± 0.021 | 0.025 ± 0.013 | -0.150 ± 0.012 | -0.800 ± 0.014 | -0.977 ± 0.007 | 2.400 ± 0.685 |
| | VNNGP | 3.005 ± 0.041 | -0.079 ± 0.010 | 0.639 ± 0.045 | 0.104 ± 0.009 | -0.056 ± 0.009 | -0.489 ± 0.008 | -0.456 ± 0.006 | 3.286 ± 0.881 |
| | CaGP | 2.794 ± 0.045 | -0.042 ± 0.022 | 0.798 ± 0.021 | 0.034 ± 0.017 | -0.103 ± 0.012 | -0.718 ± 0.013 | -0.900 ± 0.009 | 3.286 ± 0.564 |
| Test NLL | LKGP | 2.349 ± 0.047 | **0.051 ± 0.026** | 1.047 ± 0.029 | **-0.057 ± 0.019** | **-0.328 ± 0.023** | **-0.633 ± 0.029** | **-1.214 ± 0.020** | **1.257 ± 0.602** |
| | SVGP | **2.327 ± 0.065** | 0.319 ± 0.018 | **0.781 ± 0.020** | 0.199 ± 0.016 | 0.160 ± 0.018 | 0.065 ± 0.032 | -0.496 ± 0.014 | 2.543 ± 1.024 |
| | VNNGP | 2.733 ± 0.031 | 1.046 ± 0.017 | 1.365 ± 0.019 | 1.011 ± 0.016 | 0.891 ± 0.013 | 0.717 ± 0.013 | 0.712 ± 0.009 | 3.200 ± 1.141 |
| | CaGP | 2.451 ± 0.033 | 0.321 ± 0.021 | 0.916 ± 0.023 | 0.142 ± 0.016 | 0.171 ± 0.013 | 0.088 ± 0.031 | -0.463 ± 0.013 | 3.000 ± 0.000 |
| Time in min | LKGP | **0.371 ± 0.008** | **1.458 ± 0.013** | **0.487 ± 0.003** | **1.728 ± 0.010** | **2.447 ± 0.022** | **1.493 ± 0.017** | **2.012 ± 0.018** | **1.000 ± 0.000** |
| | SVGP | 6.473 ± 0.032 | 6.475 ± 0.032 | 6.479 ± 0.031 | 6.474 ± 0.032 | 6.492 ± 0.033 | 6.500 ± 0.030 | 6.475 ± 0.032 | 2.000 ± 0.000 |
| | VNNGP | 26.34 ± 0.443 | 25.89 ± 0.161 | 25.78 ± 0.151 | 25.85 ± 0.158 | 25.87 ± 0.181 | 25.85 ± 0.154 | 26.31 ± 0.464 | 4.000 ± 0.000 |
| | CaGP | 7.067 ± 0.028 | 7.036 ± 0.028 | 7.033 ± 0.023 | 7.015 ± 0.020 | 7.046 ± 0.024 | 7.024 ± 0.019 | 7.023 ± 0.020 | 3.000 ± 0.000 |

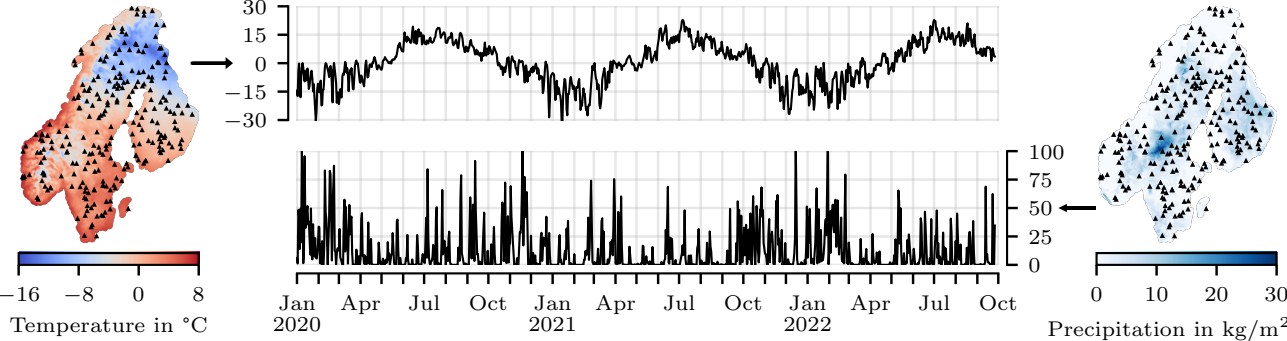

Figure 5. Illustration of daily temperature and precipitation data from the Nordic Gridded Climate Dataset (Tveito et al., 2000; 2005). The heatmaps (left and right) visualize snapshots of a single day and subsampled spatial locations. Every spatial location is associated with its own time series (middle). Temperatures (top) exhibit a seasonal periodic trend. Precipitation (bottom) is noisy but locally correlated.

without latent Kronecker structure. In the second and third experiment, we compare our method to various sparse and variational methods.

For all our experiments, we start with a gridded dataset and introduce missing values which are withheld during training and used as test data, which provides ground truth information for missing values. We compute predictive means and variances of LKGP using 64 posterior samples obtained via pathwise conditioning, akin to Lin et al. (2023). All methods are implemented using GPyTorch (Gardner et al., 2018a), and the source code is available at https://github.com/jandylin/Latent-Kronecker-GPs. All ex-

periments were conducted on A100 GPUs with a total compute time of around 2000 hours.

**Inverse Dynamics Prediction** In this experiment, we predict the inverse dynamics of a SARCOS anthropomorphic robot arm with seven degrees of freedom. The SARCOS dataset is publicly available.[1] In this problem, we wish to learn the mapping of seven joint positions, velocities, and accelerations to their corresponding seven joint torques. The main goal of this experiment is to compare LKGPs to standard iterative methods, because the former effectively

---

[1]http://gaussianprocess.org/gpml/data/

implements the latter using more efficient matrix algebra. Additionally, we empirically validate the break-even points predicted by Proposition 3.1, at which both methods should require the same amount of time or memory.

For this experiment, we consider the positions, velocities, and accelerations as $\mathcal{S}$ and the seven torques as $\mathcal{T}$. We choose $k_{\mathcal{S}}$ to be a squared exponential kernel and $k_{\mathcal{T}}$ to be a full-rank ICM kernel (Bonilla et al., 2007), demonstrating that LKGP is compatible with discrete kernels. We select subsets of the training data split with $p = 5000$ joint positions, velocities, and accelerations, and their corresponding $q = 7$ joint torques, and introduced 10%, 20%, ..., 90% missing values uniformly at random, resulting in $n \leq 35k$. The value of $p$ was chosen such that the kernel matrix without factorization fits into GPU memory. This results in a fairer comparison since it allows both methods to amortize kernel evaluation costs, such that the comparison is essentially between matrix multiplication costs (larger $p$ would force regular iterative methods to recompute kernel values, leading to even higher overall compute times). Details on the model fitting setup are provided in Appendix C.

Figure 3 illustrates the required computational resources and predictive performances for different missing data ratios. For small missing ratios, LKGP requires significantly less time and memory. As the missing ratio increases, the iterative methods eventually become slightly more efficient. Notably, the empirical break-even points, where time and memory usage is the same for both methods, matches the asymptotic break-even points predicted by Proposition 3.1. Moreover, the predictive performance of both methods is equivalent across all missing ratios in terms of test root-mean-square-error and test negative log-likelihood, validating that LKGP does not introduce any model approximation.

**Learning Curve Prediction**    In this experiment, we predict the continuation of partially observed learning curves, given some fully observed and other partially observed examples. We use data from LCBench (Zimmer et al., 2021).[2]

LCBench contains 35 distinct learning curve datasets, each containing 2000 learning curves with 52 steps each, where every step refers to a neural network training epoch. Each learning curve within a particular dataset is obtained by training a neural network on the same dataset but using different hyperparameter configurations. These hyperparameters are batch size, learning rate, momentum, weight decay, number of layers, hidden units per layer, and dropout ratio.

The main goal of this experiment is to evaluate the performance under a realistic non-uniform pattern of missing values. In particular, learning curves are observed until

a particular time step and missing all remaining values. This simulates neural network hyperparameter optimization, where learning curve information becomes available as the neural network continues to train. In this setting, learning curve predictions can be leveraged to save computational resources by pruning runs which are predicted to perform poorly (Elsken et al., 2019). We compare our method to SVGP (Hensman et al., 2013), VNNGP (Wu et al., 2022), and CaGP (Wenger et al., 2024). See Section 2 for details.

For this experiment, we consider hyperparameter configurations as $\mathcal{S}$ and learning curve progression steps as $\mathcal{T}$, such that $p = 2000$, $q = 52$, and thus $n \leq 104000$ for each dataset. We choose both $k_{\mathcal{S}}$ and $k_{\mathcal{T}}$ to be squared exponential kernels. Out of the $p = 2000$ curves per dataset, 10% were provided as fully observed during training and the remaining 90% were partially observed. The stopping point was chosen uniformly at random. See Figure 4 for an illustration and qualitative comparison of the GP methods which we considered for this experiment. Details on the model fitting setup are provided in Appendix C.

Table 1 reports the predictive performance and total runtime for every fifth LCBench dataset (see Appendix B for results on all datasets). On average, SVGP and CaGP achieve better test root-mean-square-error (RMSE), but LKGP produces the best test negative log-likelihood (NLL), providing quantitative evidence for the qualitative observation in Figure 4 which suggests that LKGP produces the most sensible uncertainty estimates.

To explain why LKGP performs worse in terms of test RMSE, we suggest that LKGP might be overfitting. Since missing values are intentionally accumulated at the end of individual learning curves to simulate early stopping, the train (observed) and test (missing) data do not have the same distribution, which makes this setting particularly prone to overfitting. Our hypothesis is empirically supported by observing that LKGP consistently achieves the best performance on the training data. The fact that LKGP still achieves the best test NLL is likely due to superior uncertainty quantification of the exact GP model compared to variational approximations, which is a commonly observed phenomenon. Furthermore, LKGP requires the least amount of time by a large margin, suggesting that it could potentially be scaled to much larger learning curve datasets.

**Climate Data with Missing Values**    In this experiment, we predict the daily average temperature and precipitation at different spatial locations. We used the Nordic Gridded Climate Dataset (Tveito et al., 2000; 2005).[3] This dataset contains observations which are gridded in space and time,

---

[2]https://github.com/automl/LCBench  available under the Apache License, Version 2.0

[3]https://cds.climate.copernicus.eu/datasets/insitu-gridded-observations-nordic available under the License to Use Copernicus Products

*Table 2.* [Climate Data with Missing Values] Predictive performances and total runtimes of temperature and precipitation prediction across various missing data ratios. LKGP consistently achieves the best predictive performance and also requires the least time across both datasets and all missing ratios. Reported numbers are the mean over 5 random seeds. Results of the best model are boldfaced and results of the second best model are underlined.

| | Model | Temperature Dataset (with Missing Ratio 10% − 50%) | | | | | Precipitation Dataset (with Missing Ratio 10% − 50%) | | | | |
|---|---|---|---|---|---|---|---|---|---|---|---|
| | | 10% | 20% | 30% | 40% | 50% | 10% | 20% | 30% | 40% | 50% |
| Train RMSE | LKGP | **0.06 ± 0.00** | **0.06 ± 0.00** | **0.07 ± 0.00** | **0.07 ± 0.00** | **0.07 ± 0.00** | **0.16 ± 0.00** | **0.17 ± 0.00** | **0.17 ± 0.00** | **0.18 ± 0.00** | **0.18 ± 0.00** |
| | SVGP | 0.21 ± 0.00 | 0.21 ± 0.00 | 0.22 ± 0.00 | 0.22 ± 0.00 | 0.23 ± 0.00 | 0.70 ± 0.00 | 0.71 ± 0.00 | 0.71 ± 0.00 | 0.71 ± 0.00 | 0.72 ± 0.00 |
| | VNNGP | 0.13 ± 0.00 | 0.13 ± 0.00 | 0.13 ± 0.00 | 0.13 ± 0.00 | 0.13 ± 0.00 | 0.45 ± 0.00 | 0.47 ± 0.00 | 0.49 ± 0.00 | 0.52 ± 0.00 | 0.58 ± 0.00 |
| | CaGP | 0.18 ± 0.00 | 0.19 ± 0.00 | 0.19 ± 0.00 | 0.19 ± 0.00 | 0.19 ± 0.00 | 0.60 ± 0.00 | 0.61 ± 0.00 | 0.61 ± 0.00 | 0.62 ± 0.00 | 0.62 ± 0.00 |
| Test RMSE | LKGP | **0.08 ± 0.00** | **0.08 ± 0.00** | **0.08 ± 0.00** | **0.09 ± 0.00** | **0.09 ± 0.00** | **0.24 ± 0.00** | **0.25 ± 0.00** | **0.26 ± 0.00** | **0.27 ± 0.00** | **0.28 ± 0.00** |
| | SVGP | 0.21 ± 0.00 | 0.21 ± 0.00 | 0.22 ± 0.00 | 0.22 ± 0.00 | 0.23 ± 0.00 | 0.70 ± 0.00 | 0.71 ± 0.00 | 0.71 ± 0.00 | 0.72 ± 0.00 | 0.72 ± 0.00 |
| | VNNGP | 0.13 ± 0.00 | 0.13 ± 0.00 | 0.13 ± 0.00 | 0.13 ± 0.00 | 0.13 ± 0.00 | 0.46 ± 0.00 | 0.48 ± 0.00 | 0.50 ± 0.00 | 0.53 ± 0.00 | 0.58 ± 0.00 |
| | CaGP | 0.18 ± 0.00 | 0.19 ± 0.00 | 0.19 ± 0.00 | 0.19 ± 0.00 | 0.19 ± 0.00 | 0.61 ± 0.00 | 0.61 ± 0.00 | 0.61 ± 0.00 | 0.62 ± 0.00 | 0.63 ± 0.00 |
| Train NLL | LKGP | **-1.33 ± 0.02** | **-1.30 ± 0.01** | **-1.26 ± 0.00** | **-1.21 ± 0.00** | **-1.18 ± 0.00** | **-0.32 ± 0.00** | **-0.29 ± 0.00** | **-0.26 ± 0.00** | **-0.23 ± 0.00** | **-0.18 ± 0.00** |
| | SVGP | -0.14 ± 0.00 | -0.12 ± 0.00 | -0.10 ± 0.00 | -0.07 ± 0.00 | -0.05 ± 0.00 | 1.07 ± 0.00 | 1.07 ± 0.00 | 1.08 ± 0.00 | 1.09 ± 0.00 | 1.10 ± 0.00 |
| | VNNGP | -0.58 ± 0.00 | -0.58 ± 0.00 | -0.57 ± 0.00 | -0.57 ± 0.00 | -0.57 ± 0.00 | 0.64 ± 0.00 | 0.67 ± 0.00 | 0.71 ± 0.00 | 0.78 ± 0.00 | 0.87 ± 0.00 |
| | CaGP | -0.22 ± 0.00 | -0.21 ± 0.00 | -0.20 ± 0.00 | -0.18 ± 0.00 | -0.17 ± 0.00 | 0.93 ± 0.00 | 0.94 ± 0.00 | 0.94 ± 0.00 | 0.95 ± 0.00 | 0.96 ± 0.00 |
| Test NLL | LKGP | **-1.14 ± 0.01** | **-1.11 ± 0.01** | **-1.07 ± 0.00** | **-1.03 ± 0.00** | **-0.99 ± 0.00** | **-0.02 ± 0.00** | **0.01 ± 0.00** | **0.04 ± 0.00** | **0.09 ± 0.00** | **0.14 ± 0.00** |
| | SVGP | -0.14 ± 0.00 | -0.12 ± 0.00 | -0.09 ± 0.00 | -0.07 ± 0.00 | -0.05 ± 0.00 | 1.07 ± 0.00 | 1.08 ± 0.00 | 1.08 ± 0.00 | 1.09 ± 0.00 | 1.10 ± 0.00 |
| | VNNGP | -0.57 ± 0.00 | -0.56 ± 0.00 | -0.56 ± 0.00 | -0.56 ± 0.00 | -0.55 ± 0.00 | 0.66 ± 0.00 | 0.69 ± 0.00 | 0.73 ± 0.00 | 0.79 ± 0.00 | 0.88 ± 0.00 |
| | CaGP | -0.22 ± 0.00 | -0.20 ± 0.00 | -0.19 ± 0.00 | -0.18 ± 0.00 | -0.17 ± 0.00 | 0.94 ± 0.00 | 0.94 ± 0.00 | 0.94 ± 0.00 | 0.95 ± 0.00 | 0.96 ± 0.00 |
| Time in min | LKGP | **28.7 ± 0.33** | **26.9 ± 0.41** | **24.6 ± 0.19** | **23.0 ± 0.31** | **21.2 ± 0.16** | **15.6 ± 0.01** | **14.2 ± 0.01** | **12.7 ± 0.01** | **11.3 ± 0.00** | **9.87 ± 0.01** |
| | SVGP | 150 ± 0.08 | 133 ± 0.07 | 117 ± 0.04 | 100 ± 0.03 | 83.3 ± 0.03 | 150 ± 0.02 | 133 ± 0.13 | 116 ± 0.02 | 99.9 ± 0.05 | 83.2 ± 0.03 |
| | VNNGP | 158 ± 0.26 | 140 ± 0.17 | 123 ± 0.37 | 105 ± 0.16 | 87.8 ± 0.13 | 158 ± 0.40 | 141 ± 0.27 | 123 ± 0.11 | 105 ± 0.13 | 87.6 ± 0.22 |
| | CaGP | 500 ± 0.15 | 395 ± 0.06 | 301 ± 0.12 | 221 ± 0.01 | 153 ± 0.01 | 500 ± 0.02 | 395 ± 0.03 | 301 ± 0.18 | 221 ± 0.06 | 153 ± 0.07 |

namely latitude, longitude and days. The main goal of this experiment is to demonstrate the scalability of LKGP on large datasets with millions of observations. Additionally, we investigated how increasing the missing data ratio impacts predictive performances and runtime requirements. We compare our method to the same sparse and variational methods from the previous experiment.

For this experiment, we considered latitude and longitude as $\mathcal{S}$ and time in days as $\mathcal{T}$. We selected $p = 5000$ spatial locations uniformly at random together with $q = 1000$ days of temperature or precipitation measurements, starting on January 1st, 2020, resulting in two datasets of size $n \leq 5M$. See Figure 5 for an illustration. We choose $k_{\mathcal{S}}$ to be a squared exponential kernel and $k_{\mathcal{T}}$ to be the product of a squared exponential kernel and a periodic kernel to capture seasonal trends. Missing values were selected uniformly at random with ratios of 10%, 20%, ..., 50%. Details on the model fitting setup are provided in Appendix C.

Table 2 reports the predictive performances and total runtimes of temperature and precipitation prediction. On this large dataset with millions of examples, LKGP clearly outperforms the other sparse and variational methods in both root-mean-square-error and negative log-likelihood. This is unsurprising, given that LKGP performs inference in the exact GP model while the other methods are subject to a limited number of inducing points, nearest neighbors, or sparse actions. However, due to latent Kronecker structure, LKGP also requires the least amount of time by a large margin. Interestingly, VNNGP consistently outperforms SVGP

and CaGP on this problem. We suspect that this is due to the nearest neighbor mechanism working well on these datasets with actual spatial dimensions.

## 5. Conclusion

We proposed a highly scalable exact Gaussian process model for product kernels, which leverages latent Kronecker structure to accelerate computations and reduce memory requirements for data arranged on a partial grid while allowing for missing values. In contrast to existing Gaussian process models with Kronecker structure, our approach deals with missing values by combining projections and iterative linear algebra methods. Empirically, we demonstrated that our method has superior computational scalability compared to the standard iterative methods, and substantially outperforms sparse variational GPs in terms of prediction quality. Future work could investigate specialized kernels, multi-product generalizations, and heteroskedastic noise models.

**Limitations** The primary limitation of LKGP is that it employs a product kernel, which assumes that points are only correlated if they are close in both $\mathcal{S}$ and $\mathcal{T}$. This is a reasonable assumption in many settings, such as correlated time series data, but will not always be appropriate. The other requirement for our approach is that the data lives on a partial grid, which is less restrictive since our model is competitive even if there are a lot of missing values, as our experiments and theory show. Furthermore, if the data does not even live on a partial grid, it would be possible to generate an artificial grid, for example via local interpolation.

## Impact Statement

This paper presents work whose goal is to advance the field of Machine Learning. There are many potential societal consequences of our work, none which we feel must be specifically highlighted here.

## Acknowledgments

J. A. Lin was supported by the University of Cambridge Harding Distinguished Postgraduate Scholars Programme, and J. M. Hernández-Lobato acknowledges support from a Turing AI Fellowship under grant EP/V023756/1. This work was performed using computational resources provided by the Cambridge Tier-2 system operated by the University of Cambridge Research Computing Service.

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

## A. Derivation of Asymptotic Break-even Points

To derive the asymptotic break-even points for MVM time and memory usage, as stated in Proposition 3.1, we start by defining the missing ratio $\gamma = 1 - n/pq$. Alternatively, this can also be written as $n = (1 - \gamma)pq$. Furthermore, we have the properties $n, p, q > 0$ and $n \leq pq$, and thus $\gamma \in [0, 1)$.

The asymptotic time complexity of MVM of a $n \times n$ matrix with a vector of $n$ elements is $\mathcal{O}(n^2)$. For a matrix which can be factorized as the Kronecker product of two matrices of sizes $p \times p$ and $q \times q$ respectively, the asymptotic time complexity of MVM is $\mathcal{O}(p^2 q + p q^2)$. To find the asymptotic break-even point, we equate these asymptotic time complexities and write $n$ in terms of $\gamma$, $p$, and $q$,

$$n^2 = p^2 q + p q^2,$$
$$((1 - \gamma)pq)^2 = p^2 q + p q^2,$$
$$(1 - \gamma)^2 p^2 q^2 = p^2 q + p q^2,$$
$$(1 - \gamma)^2 = \frac{1}{q} + \frac{1}{p}.$$

Since $\gamma$ is the missing ratio, which is between $0$ and $1$, we conclude that the asymptotic break-even point for time is

$$\gamma_{\text{time}}^* = 1 - \sqrt{\frac{1}{p} + \frac{1}{q}}.$$

For the asymptotic space complexity, we follow the same approach, starting with $\mathcal{O}(n^2)$ and $\mathcal{O}(p^2 + q^2)$. Again, we set these terms equal to each other and express $n$ in terms of $\gamma$, $p$, and $q$,

$$n^2 = p^2 + q^2,$$
$$((1 - \gamma)pq)^2 = p^2 + q^2,$$
$$(1 - \gamma)^2 p^2 q^2 = p^2 + q^2,$$
$$(1 - \gamma)^2 = \frac{1}{q^2} + \frac{1}{p^2}.$$

We conclude that the asymptotic break-even point for memory is

$$\gamma_{\text{mem}}^* = 1 - \sqrt{\frac{1}{p^2} + \frac{1}{q^2}}.$$

$\square$

Since the asymptotic time and space complexity of kernel evaluations is equivalent to the case of asymptotic MVM memory usage with $\mathcal{O}(n^2)$ and $\mathcal{O}(p^2 + q^2)$, we conclude that the derivation and result of the second case can be equivalently applied to reason about the asymptotic break-even points of kernel evaluation time and memory.

## B. Additional Experiment Results

Tables 3 to 7 report the predictive performances and total runtimes of our learning curve prediction experiment on all LCBench datasets. See Section 4 for details about the experiment.

*Table 3.* Predictive performances and total runtimes of learning curve prediction on LCBench datasets $1 - 7$. Reported numbers are the mean over 10 random seeds. Results of the best model are boldfaced and results of the second best model are underlined.

| | Model | APSFailure | Amazon | Australian | Fashion | KDDCup09 | MiniBooNE | adult |
|---|---|---|---|---|---|---|---|---|
| **Train RMSE** | LKGP | **1.705 ± 0.047** | 0.273 ± 0.005 | **0.007 ± 0.000** | **0.094 ± 0.006** | **1.609 ± 0.233** | **0.047 ± 0.001** | **0.053 ± 0.001** |
| | SVGP | 3.893 ± 0.149 | 0.402 ± 0.007 | 0.108 ± 0.001 | 0.362 ± 0.017 | 1.775 ± 0.247 | 0.201 ± 0.004 | 0.244 ± 0.004 |
| | VNNGP | 4.920 ± 0.194 | **0.203 ± 0.003** | 0.076 ± 0.001 | 0.341 ± 0.174 | 3.194 ± 0.490 | 0.113 ± 0.001 | 0.131 ± 0.002 |
| | CaGP | 3.972 ± 0.147 | 0.417 ± 0.007 | 0.119 ± 0.001 | 0.319 ± 0.015 | 1.839 ± 0.262 | 0.219 ± 0.005 | 0.265 ± 0.005 |
| **Test RMSE** | LKGP | 2.935 ± 0.150 | 0.386 ± 0.007 | 0.211 ± 0.005 | 0.597 ± 0.081 | 2.080 ± 0.360 | 0.335 ± 0.006 | 0.390 ± 0.008 |
| | SVGP | **2.716 ± 0.145** | 0.311 ± 0.006 | 0.195 ± 0.001 | 0.611 ± 0.046 | **2.000 ± 0.330** | **0.316 ± 0.005** | **0.351 ± 0.005** |
| | VNNGP | 3.053 ± 0.159 | 0.663 ± 0.015 | 0.593 ± 0.005 | 0.945 ± 0.074 | 2.857 ± 0.467 | 0.705 ± 0.013 | 0.713 ± 0.010 |
| | CaGP | 2.810 ± 0.154 | **0.299 ± 0.006** | **0.193 ± 0.002** | **0.541 ± 0.046** | 2.058 ± 0.331 | 0.325 ± 0.006 | 0.358 ± 0.006 |
| **Train NLL** | LKGP | **1.955 ± 0.020** | **0.122 ± 0.019** | -2.924 ± 0.005 | **-0.918 ± 0.058** | 1.796 ± 0.157 | **-1.573 ± 0.022** | **-1.454 ± 0.021** |
| | SVGP | 3.098 ± 0.100 | 0.499 ± 0.017 | -0.746 ± 0.010 | 0.410 ± 0.047 | **1.749 ± 0.128** | -0.132 ± 0.020 | 0.050 ± 0.016 |
| | VNNGP | 3.005 ± 0.041 | 0.302 ± 0.013 | -0.418 ± 0.005 | 0.373 ± 0.189 | 2.467 ± 0.154 | -0.079 ± 0.010 | 0.033 ± 0.010 |
| | CaGP | 2.794 ± 0.045 | 0.524 ± 0.017 | -0.648 ± 0.011 | 0.313 ± 0.047 | 1.927 ± 0.145 | -0.042 ± 0.022 | 0.141 ± 0.018 |
| **Test NLL** | LKGP | 2.349 ± 0.047 | 0.455 ± 0.018 | **-0.798 ± 0.013** | 0.349 ± 0.066 | 1.931 ± 0.195 | **0.051 ± 0.026** | **0.146 ± 0.009** |
| | SVGP | **2.327 ± 0.065** | 0.370 ± 0.018 | -0.190 ± 0.012 | 0.766 ± 0.063 | **1.923 ± 0.172** | 0.319 ± 0.018 | 0.401 ± 0.016 |
| | VNNGP | 2.733 ± 0.031 | 1.019 ± 0.018 | 0.753 ± 0.007 | 1.385 ± 0.071 | 2.387 ± 0.150 | 1.046 ± 0.017 | 1.086 ± 0.012 |
| | CaGP | 2.451 ± 0.033 | **0.355 ± 0.017** | -0.221 ± 0.011 | 0.723 ± 0.123 | 1.999 ± 0.149 | 0.321 ± 0.021 | 0.404 ± 0.017 |
| **Time in min** | LKGP | **0.371 ± 0.008** | **0.620 ± 0.003** | **1.680 ± 0.014** | **1.435 ± 0.051** | **0.407 ± 0.013** | **1.458 ± 0.013** | **1.539 ± 0.011** |
| | SVGP | 6.473 ± 0.032 | 6.472 ± 0.032 | 6.475 ± 0.033 | 6.472 ± 0.030 | 6.471 ± 0.032 | 6.475 ± 0.032 | 6.472 ± 0.032 |
| | VNNGP | 26.34 ± 0.443 | 25.82 ± 0.161 | 25.95 ± 0.151 | 25.84 ± 0.171 | 25.77 ± 0.140 | 25.89 ± 0.161 | 25.93 ± 0.132 |
| | CaGP | 7.067 ± 0.028 | 7.025 ± 0.021 | 7.054 ± 0.025 | 7.027 ± 0.020 | 7.084 ± 0.022 | 7.036 ± 0.028 | 7.070 ± 0.026 |

*Table 4.* Predictive performances and total runtimes of learning curve prediction on LCBench datasets $8 - 14$. Reported numbers are the mean over 10 random seeds. Results of the best model are boldfaced and results of the second best model are underlined.

| | Model | airlines | albert | bank | blood | car | christine | cnae-9 |
|---|---|---|---|---|---|---|---|---|
| **Train RMSE** | LKGP | **0.080 ± 0.001** | **0.055 ± 0.002** | **0.122 ± 0.002** | **0.278 ± 0.017** | **0.017 ± 0.000** | **0.947 ± 0.112** | **0.005 ± 0.000** |
| | SVGP | 0.293 ± 0.004 | 0.229 ± 0.002 | 0.261 ± 0.003 | 0.463 ± 0.009 | 0.111 ± 0.001 | 1.229 ± 0.135 | 0.075 ± 0.001 |
| | VNNGP | 0.168 ± 0.002 | 0.121 ± 0.001 | 0.135 ± 0.001 | 0.284 ± 0.029 | 0.080 ± 0.001 | 4.379 ± 0.480 | 0.059 ± 0.000 |
| | CaGP | 0.310 ± 0.004 | 0.257 ± 0.002 | 0.284 ± 0.004 | 0.511 ± 0.011 | 0.128 ± 0.001 | 1.409 ± 0.173 | 0.074 ± 0.001 |
| **Test RMSE** | LKGP | 0.377 ± 0.008 | 0.399 ± 0.019 | 0.371 ± 0.008 | 0.747 ± 0.016 | 0.265 ± 0.008 | 2.829 ± 0.487 | 0.188 ± 0.004 |
| | SVGP | 0.259 ± 0.004 | **0.308 ± 0.005** | **0.335 ± 0.005** | **0.540 ± 0.013** | **0.199 ± 0.003** | 2.570 ± 0.408 | **0.166 ± 0.002** |
| | VNNGP | 0.532 ± 0.008 | 0.548 ± 0.007 | 0.712 ± 0.009 | 0.915 ± 0.017 | 0.595 ± 0.004 | 4.600 ± 0.658 | 0.555 ± 0.006 |
| | CaGP | **0.255 ± 0.004** | 0.321 ± 0.004 | 0.341 ± 0.006 | 0.593 ± 0.015 | 0.204 ± 0.002 | 3.152 ± 0.515 | 0.176 ± 0.003 |
| **Train NLL** | LKGP | **-1.051 ± 0.015** | **-1.412 ± 0.042** | **-0.664 ± 0.016** | 0.160 ± 0.055 | **-2.528 ± 0.010** | 1.229 ± 0.200 | **-3.042 ± 0.003** |
| | SVGP | 0.181 ± 0.012 | -0.031 ± 0.009 | 0.093 ± 0.013 | 0.652 ± 0.021 | -0.713 ± 0.006 | 1.389 ± 0.162 | -1.113 ± 0.010 |
| | VNNGP | 0.065 ± 0.010 | -0.116 ± 0.005 | 0.053 ± 0.006 | 0.639 ± 0.045 | -0.382 ± 0.007 | 2.788 ± 0.176 | -0.620 ± 0.007 |
| | CaGP | 0.222 ± 0.011 | 0.095 ± 0.007 | 0.177 ± 0.014 | 0.798 ± 0.021 | -0.577 ± 0.007 | 1.709 ± 0.190 | -1.084 ± 0.010 |
| **Test NLL** | LKGP | 0.349 ± 0.013 | **0.018 ± 0.025** | **0.231 ± 0.014** | 1.047 ± 0.029 | **-0.537 ± 0.018** | 2.425 ± 0.368 | **-0.863 ± 0.048** |
| | SVGP | **0.137 ± 0.011** | 0.215 ± 0.014 | 0.329 ± 0.015 | **0.781 ± 0.020** | -0.173 ± 0.017 | **2.140 ± 0.278** | -0.349 ± 0.025 |
| | VNNGP | 0.736 ± 0.012 | 0.772 ± 0.007 | 1.072 ± 0.013 | 1.365 ± 0.019 | 0.763 ± 0.007 | 2.852 ± 0.220 | 0.587 ± 0.010 |
| | CaGP | 0.137 ± 0.011 | 0.266 ± 0.010 | 0.344 ± 0.015 | 0.916 ± 0.023 | -0.174 ± 0.013 | 2.538 ± 0.332 | -0.242 ± 0.034 |
| **Time in min** | LKGP | **1.182 ± 0.008** | **1.937 ± 0.044** | **0.908 ± 0.006** | **0.487 ± 0.003** | **1.341 ± 0.007** | **0.459 ± 0.063** | **1.531 ± 0.010** |
| | SVGP | 6.479 ± 0.035 | 6.472 ± 0.032 | 6.481 ± 0.036 | 6.479 ± 0.031 | 6.474 ± 0.030 | 6.469 ± 0.031 | 6.495 ± 0.033 |
| | VNNGP | 25.92 ± 0.170 | 25.87 ± 0.140 | 25.81 ± 0.156 | 25.78 ± 0.151 | 25.88 ± 0.143 | 25.83 ± 0.152 | 26.39 ± 0.471 |
| | CaGP | 7.129 ± 0.044 | 7.034 ± 0.020 | 7.019 ± 0.028 | 7.033 ± 0.023 | 7.042 ± 0.023 | 7.019 ± 0.020 | 7.032 ± 0.019 |

*Table 5.* Predictive performances and total runtimes of learning curve prediction on LCBench datasets $15 - 21$. Reported numbers are the mean over 10 random seeds. Results of the best model are boldfaced and results of the second best model are underlined.

| | Model | connect-4 | covertype | credit-g | dionis | fabert | helena | higgs |
|---|---|---|---|---|---|---|---|---|
| **Train RMSE** | LKGP | 0.149 ± 0.005 | **0.061 ± 0.001** | **0.047 ± 0.000** | **0.008 ± 0.000** | **0.027 ± 0.001** | **0.012 ± 0.000** | **0.039 ± 0.001** |
| | SVGP | 0.239 ± 0.009 | 0.240 ± 0.003 | 0.133 ± 0.001 | 0.114 ± 0.003 | 0.116 ± 0.001 | 0.108 ± 0.001 | 0.200 ± 0.003 |
| | VNNGP | **0.117 ± 0.003** | 0.158 ± 0.003 | 0.123 ± 0.002 | 0.088 ± 0.002 | 0.072 ± 0.001 | 0.088 ± 0.001 | 0.130 ± 0.002 |
| | CaGP | 0.265 ± 0.011 | 0.241 ± 0.004 | 0.174 ± 0.002 | 0.123 ± 0.003 | 0.119 ± 0.001 | 0.122 ± 0.001 | 0.208 ± 0.002 |
| **Test RMSE** | LKGP | 0.400 ± 0.021 | 0.351 ± 0.008 | 0.315 ± 0.008 | **0.249 ± 0.008** | 0.276 ± 0.007 | 0.257 ± 0.030 | 0.394 ± 0.027 |
| | SVGP | **0.363 ± 0.021** | 0.294 ± 0.005 | **0.257 ± 0.004** | 0.264 ± 0.010 | **0.230 ± 0.004** | **0.189 ± 0.004** | **0.285 ± 0.005** |
| | VNNGP | 0.722 ± 0.034 | 0.677 ± 0.011 | 0.585 ± 0.002 | 0.646 ± 0.015 | 0.613 ± 0.007 | 0.619 ± 0.010 | 0.642 ± 0.009 |
| | CaGP | 0.388 ± 0.022 | **0.281 ± 0.004** | 0.270 ± 0.002 | 0.254 ± 0.009 | 0.243 ± 0.003 | 0.205 ± 0.005 | 0.296 ± 0.004 |
| **Train NLL** | LKGP | **-0.482 ± 0.034** | **-1.311 ± 0.017** | **-1.611 ± 0.010** | **-2.941 ± 0.010** | **-2.115 ± 0.052** | **-2.727 ± 0.019** | **-1.718 ± 0.018** |
| | SVGP | -0.027 ± 0.038 | 0.025 ± 0.013 | -0.515 ± 0.006 | -0.687 ± 0.021 | -0.672 ± 0.010 | -0.745 ± 0.011 | -0.150 ± 0.012 |
| | VNNGP | -0.101 ± 0.024 | 0.104 ± 0.009 | -0.132 ± 0.010 | -0.334 ± 0.016 | -0.429 ± 0.006 | -0.323 ± 0.008 | -0.056 ± 0.009 |
| | CaGP | 0.086 ± 0.040 | 0.034 ± 0.017 | -0.265 ± 0.009 | -0.557 ± 0.022 | -0.609 ± 0.011 | -0.614 ± 0.012 | -0.103 ± 0.012 |
| **Test NLL** | LKGP | **0.132 ± 0.048** | **-0.057 ± 0.019** | **-0.202 ± 0.018** | **-0.551 ± 0.033** | **-0.329 ± 0.060** | **-0.658 ± 0.054** | **-0.328 ± 0.023** |
| | SVGP | 0.267 ± 0.053 | 0.199 ± 0.016 | 0.018 ± 0.020 | 0.209 ± 0.060 | -0.044 ± 0.024 | -0.215 ± 0.028 | 0.160 ± 0.018 |
| | VNNGP | 0.958 ± 0.047 | 1.011 ± 0.016 | 0.801 ± 0.005 | 0.880 ± 0.024 | 0.771 ± 0.009 | 0.826 ± 0.014 | 0.891 ± 0.013 |
| | CaGP | 0.421 ± 0.061 | 0.142 ± 0.016 | 0.113 ± 0.010 | 0.182 ± 0.062 | 0.018 ± 0.025 | -0.155 ± 0.028 | 0.171 ± 0.013 |
| **Time in min** | LKGP | **0.679 ± 0.003** | **1.728 ± 0.010** | **0.925 ± 0.009** | **1.622 ± 0.020** | **1.432 ± 0.018** | **2.066 ± 0.022** | **2.447 ± 0.022** |
| | SVGP | 6.480 ± 0.032 | 6.474 ± 0.032 | 6.493 ± 0.036 | 6.476 ± 0.031 | 6.493 ± 0.031 | 6.476 ± 0.035 | 6.492 ± 0.033 |
| | VNNGP | 25.89 ± 0.181 | 25.85 ± 0.158 | 25.84 ± 0.151 | 26.34 ± 0.512 | 25.91 ± 0.155 | 25.86 ± 0.167 | 25.87 ± 0.181 |
| | CaGP | 7.081 ± 0.029 | 7.015 ± 0.020 | 7.024 ± 0.021 | 7.041 ± 0.023 | 7.015 ± 0.026 | 7.018 ± 0.022 | 7.046 ± 0.024 |

*Table 6.* Predictive performances and total runtimes of learning curve prediction on LCBench datasets $22 - 28$. Reported numbers are the mean over 10 random seeds. Results of the best model are boldfaced and results of the second best model are underlined.

| | Model | jannis | jasmine | jungle | kc1 | kr-vs-kp | mfeat-factors | nomao |
|---|---|---|---|---|---|---|---|---|
| **Train RMSE** | LKGP | **0.074 ± 0.001** | **0.027 ± 0.001** | **0.044 ± 0.001** | 0.282 ± 0.006 | **0.008 ± 0.000** | **0.006 ± 0.000** | **0.027 ± 0.001** |
| | SVGP | 0.224 ± 0.004 | 0.146 ± 0.002 | 0.176 ± 0.003 | 0.408 ± 0.008 | 0.101 ± 0.001 | 0.099 ± 0.001 | 0.245 ± 0.006 |
| | VNNGP | 0.128 ± 0.001 | 0.088 ± 0.001 | 0.109 ± 0.001 | **0.185 ± 0.002** | 0.067 ± 0.001 | 0.079 ± 0.001 | 0.133 ± 0.002 |
| | CaGP | 0.243 ± 0.004 | 0.149 ± 0.001 | 0.201 ± 0.004 | 0.459 ± 0.008 | 0.105 ± 0.002 | 0.100 ± 0.001 | 0.245 ± 0.007 |
| **Test RMSE** | LKGP | 0.375 ± 0.014 | 0.290 ± 0.008 | 0.314 ± 0.005 | 0.497 ± 0.011 | 0.261 ± 0.005 | 0.165 ± 0.003 | 0.324 ± 0.015 |
| | SVGP | **0.314 ± 0.005** | **0.264 ± 0.004** | **0.287 ± 0.006** | 0.442 ± 0.013 | **0.232 ± 0.003** | **0.160 ± 0.001** | 0.302 ± 0.009 |
| | VNNGP | 0.675 ± 0.013 | 0.627 ± 0.003 | 0.683 ± 0.012 | 0.726 ± 0.007 | 0.591 ± 0.006 | 0.563 ± 0.003 | 0.702 ± 0.018 |
| | CaGP | 0.324 ± 0.006 | 0.280 ± 0.003 | 0.312 ± 0.005 | **0.428 ± 0.008** | 0.239 ± 0.004 | 0.165 ± 0.001 | **0.288 ± 0.008** |
| **Train NLL** | LKGP | **-1.138 ± 0.020** | **-2.094 ± 0.031** | **-1.647 ± 0.017** | **0.163 ± 0.023** | **-2.933 ± 0.005** | **-2.930 ± 0.005** | **-2.035 ± 0.034** |
| | SVGP | -0.045 ± 0.016 | -0.461 ± 0.009 | -0.266 ± 0.016 | 0.495 ± 0.021 | -0.800 ± 0.014 | -0.836 ± 0.010 | 0.048 ± 0.025 |
| | VNNGP | -0.023 ± 0.009 | -0.292 ± 0.007 | -0.127 ± 0.010 | 0.264 ± 0.009 | -0.489 ± 0.008 | -0.418 ± 0.009 | 0.032 ± 0.012 |
| | CaGP | 0.038 ± 0.016 | -0.414 ± 0.009 | -0.139 ± 0.017 | 0.635 ± 0.017 | -0.718 ± 0.013 | -0.802 ± 0.009 | 0.064 ± 0.028 |
| **Test NLL** | LKGP | **0.148 ± 0.025** | **-0.382 ± 0.018** | **-0.194 ± 0.011** | 0.638 ± 0.020 | **-0.633 ± 0.029** | **-1.064 ± 0.015** | **0.104 ± 0.034** |
| | SVGP | 0.272 ± 0.017 | 0.034 ± 0.012 | 0.184 ± 0.020 | **0.590 ± 0.009** | 0.065 ± 0.032 | -0.396 ± 0.013 | 0.285 ± 0.031 |
| | VNNGP | 0.995 ± 0.017 | 0.851 ± 0.006 | 0.974 ± 0.015 | 1.105 ± 0.009 | 0.717 ± 0.013 | 0.711 ± 0.006 | 1.130 ± 0.027 |
| | CaGP | 0.286 ± 0.017 | 0.131 ± 0.015 | 0.259 ± 0.017 | 0.616 ± 0.011 | 0.088 ± 0.031 | -0.388 ± 0.010 | 0.229 ± 0.030 |
| **Time in min** | LKGP | **1.327 ± 0.008** | **1.867 ± 0.016** | **1.481 ± 0.011** | **0.576 ± 0.009** | **1.493 ± 0.017** | **1.942 ± 0.018** | **2.183 ± 0.020** |
| | SVGP | 6.478 ± 0.032 | 6.495 ± 0.030 | 6.482 ± 0.031 | 6.491 ± 0.033 | 6.500 ± 0.030 | 6.491 ± 0.030 | 6.487 ± 0.032 |
| | VNNGP | 26.40 ± 0.449 | 25.91 ± 0.162 | 25.89 ± 0.121 | 25.89 ± 0.143 | 25.85 ± 0.154 | 26.34 ± 0.413 | 25.86 ± 0.167 |
| | CaGP | 7.009 ± 0.025 | 7.019 ± 0.021 | 7.034 ± 0.029 | 7.016 ± 0.024 | 7.024 ± 0.019 | 7.034 ± 0.030 | 7.033 ± 0.023 |

*Table 7.* Predictive performances and total runtimes of learning curve prediction on LCBench datasets $29 - 35$. Reported numbers are the mean over 10 random seeds. Results of the best model are boldfaced and results of the second best model are underlined.

| | Model | numerai28.6 | phoneme | segment | shuttle | sylvine | vehicle | volkert |
|---|---|---|---|---|---|---|---|---|
| **Train RMSE** | LKGP | **1.007 ± 0.156** | **0.062 ± 0.001** | **0.006 ± 0.000** | **1.172 ± 0.265** | **0.008 ± 0.000** | **0.017 ± 0.000** | **0.043 ± 0.001** |
| | SVGP | 1.669 ± 0.188 | 0.208 ± 0.002 | 0.086 ± 0.001 | 1.316 ± 0.286 | 0.109 ± 0.001 | 0.077 ± 0.001 | 0.147 ± 0.016 |
| | VNNGP | 5.900 ± 0.618 | 0.132 ± 0.001 | 0.072 ± 0.001 | 2.611 ± 0.620 | 0.075 ± 0.001 | 0.062 ± 0.001 | 0.105 ± 0.001 |
| | CaGP | 1.327 ± 0.157 | 0.238 ± 0.002 | 0.091 ± 0.001 | 1.361 ± 0.303 | 0.118 ± 0.002 | 0.089 ± 0.001 | 0.171 ± 0.003 |
| **Test RMSE** | LKGP | **0.831 ± 0.099** | 0.349 ± 0.007 | 0.152 ± 0.003 | 1.597 ± 0.368 | 0.253 ± 0.007 | 0.203 ± 0.007 | 0.260 ± 0.007 |
| | SVGP | 1.104 ± 0.120 | **0.285 ± 0.005** | **0.145 ± 0.002** | 1.568 ± 0.371 | **0.228 ± 0.003** | **0.169 ± 0.002** | **0.200 ± 0.021** |
| | VNNGP | 1.686 ± 0.180 | 0.676 ± 0.005 | 0.568 ± 0.005 | 2.429 ± 0.585 | 0.595 ± 0.006 | 0.576 ± 0.006 | 0.617 ± 0.010 |
| | CaGP | 0.882 ± 0.096 | 0.295 ± 0.004 | 0.153 ± 0.002 | **1.551 ± 0.356** | 0.241 ± 0.003 | 0.186 ± 0.003 | 0.228 ± 0.004 |
| **Train NLL** | LKGP | **1.393 ± 0.140** | **-1.323 ± 0.010** | **-2.943 ± 0.005** | 1.348 ± 0.219 | **-2.900 ± 0.008** | **-2.568 ± 0.012** | **-1.668 ± 0.029** |
| | SVGP | 1.890 ± 0.128 | -0.112 ± 0.007 | -0.977 ± 0.007 | 1.435 ± 0.205 | -0.728 ± 0.011 | -1.083 ± 0.009 | -0.321 ± 0.036 |
| | VNNGP | 3.146 ± 0.099 | 0.003 ± 0.005 | -0.456 ± 0.006 | 2.130 ± 0.221 | -0.414 ± 0.009 | -0.567 ± 0.007 | -0.193 ± 0.009 |
| | CaGP | 1.668 ± 0.105 | 0.015 ± 0.007 | -0.900 ± 0.009 | 1.490 ± 0.208 | -0.624 ± 0.014 | -0.942 ± 0.008 | -0.303 ± 0.017 |
| **Test NLL** | LKGP | **1.372 ± 0.099** | **-0.050 ± 0.017** | **-1.214 ± 0.020** | 1.533 ± 0.247 | **-0.641 ± 0.026** | **-0.923 ± 0.034** | **-0.266 ± 0.030** |
| | SVGP | 1.595 ± 0.075 | 0.168 ± 0.014 | -0.496 ± 0.014 | 1.573 ± 0.246 | -0.021 ± 0.026 | -0.381 ± 0.019 | -0.077 ± 0.022 |
| | VNNGP | 2.734 ± 0.100 | 0.991 ± 0.007 | 0.712 ± 0.009 | 2.079 ± 0.221 | 0.765 ± 0.009 | 0.624 ± 0.010 | 0.883 ± 0.016 |
| | CaGP | 1.519 ± 0.099 | 0.199 ± 0.011 | -0.463 ± 0.013 | 1.577 ± 0.219 | 0.049 ± 0.020 | -0.229 ± 0.019 | -0.084 ± 0.018 |
| **Time in min** | LKGP | **0.410 ± 0.007** | **1.136 ± 0.005** | **2.012 ± 0.018** | **0.471 ± 0.023** | **1.629 ± 0.018** | **1.368 ± 0.009** | **1.821 ± 0.018** |
| | SVGP | 6.477 ± 0.030 | 6.478 ± 0.029 | 6.475 ± 0.032 | 6.471 ± 0.031 | 6.474 ± 0.032 | 6.471 ± 0.031 | 5.827 ± 0.615 |
| | VNNGP | 25.95 ± 0.162 | 25.82 ± 0.181 | 26.31 ± 0.464 | 25.94 ± 0.171 | 25.85 ± 0.168 | 26.11 ± 0.174 | 26.33 ± 0.442 |
| | CaGP | 7.044 ± 0.026 | 7.023 ± 0.016 | 7.023 ± 0.020 | 7.033 ± 0.032 | 7.014 ± 0.024 | 7.029 ± 0.025 | 7.035 ± 0.020 |

## C. Experiment Details

**Inverse Dynamics Prediction** For both methods, observation noise and kernel hyperparameters were initialized with GPyTorch default values, and inferred by using Adam with a learning rate of 0.1 to maximize the marginal likelihood for 50 iterations. Additionally, both methods use conjugate gradients with a relative residual norm tolerance of 0.01 as iterative linear system solver.

**Learning Curve Prediction** For all methods, observation noise and kernel hyperparameters were initialized with GPyTorch default values, and optimized using Adam. LKGP was trained for 100 iterations using a learning rate of 0.1, conjugate gradients with a relative residual norm tolerance of 0.01 and a pivoted Cholesky preconditioner of rank 100. SVGP was trained for 30 epochs using a learning rate of 0.01, a batch size of 1000, and 10000 inducing points, which were initialized at random training data examples. VNNGP was trained for 1000 epochs using a learning rate of 0.01, a batch size of 1000, inducing points placed at every training example, and 256 nearest neighbors. CaGP was trained for 1000 epochs using a learning rate of 0.1 and 512 actions.

**Climate Data with Missing Values** For all methods, observation noise and kernel hyperparameters were initialized with GPyTorch default values and optimized using Adam. LKGP was trained for 100 iterations using a learning rate of 0.1, and conjugate gradients with a relative residual norm tolerance of 0.01 and a pivoted Cholesky preconditioner of rank 100. SVGP was trained for 5 epochs using a learning rate of 0.001, a batch size of 1000, and 10000 inducing points, which were initialized at random training data examples. VNNGP was trained for 50 epochs using a learning rate of 0.001, a batch size of 1000, 500000 inducing points placed at random training data examples, and 256 nearest neighbors. CaGP was trained for 50 epochs using a learning rate of 0.1 and 256 actions.

