# OpenReview forum: "Scalable Gaussian Processes with Latent Kronecker Structure"
_ICML.cc/2025/Conference — ICML 2025 poster_

### Official Review · Reviewer_1bD1 · 2025-03-11

**Overall Recommendation:** 3

**Summary:**

This paper considers a Gaussian process (GP) regression with data points on a Cartesian grid and proposes a new way of constructing GP gram matrices to deal with missing values. The proposed method represents a gram matrix with missing values as a projection of a latent exact gram matrix, enabling efficient GP inference by combining the Kronecker structure of the gram matrix with iterative methods. The experiments on synthetic and real-world data confirmed the model's validity.

**Claims And Evidence:**

The relationship to related works regarding the handling of missing values is not clearly presented. The discussion of related works appears to focus primarily on GP models without missing values.

**Essential References Not Discussed:**

The relationship to related works regarding the handling of missing values seems not to be clearly presented. For example, how is the proposed method related to [Smola et al., Kernel methods for missing variables, 2005], [Jafrasteh et al., Gaussian processes for missing value imputation, 2023], and [Imani et al, Nested Gaussian process modeling and imputation of high-dimensional incomplete data under uncertainty, 2019]?

**Experimental Designs Or Analyses:**

No issues specified.

**Methods And Evaluation Criteria:**

The proposed methods and evaluation criteria make sense.

**Other Comments Or Suggestions:**

No other comments.

**Other Strengths And Weaknesses:**

Strength
- The idea of the paper is clearly presented, and the proposed method is easy-to-implement.
- The validity of the proposed model was evaluated on a lot of real-world data.

Weakness
- See above.

**Questions For Authors:**

I cannot follow the sentence in pp. 2, " While the above methods scale gracefully to large datasets, they are all are fundamentally limited by performing inference of approximate GP models.". Why are the sparse and variational methods fundamentally limited?

**Relation To Broader Scientific Literature:**

This paper provides an easy-to-implement algorithm for GP models with missing values, which could be beneficial in many research fields because GP models have been widely utilized.

**Theoretical Claims:**

Theoretical claims seem to be correct.

---

> ### Author Rebuttal · Authors · 2025-03-28
>
> Dear Reviewer 1bD1,
>
> Thank you for your time and effort spent on reviewing our paper. In the following, we provide responses to your specific concerns and questions.
>
> ---
>
> > The relationship to related works regarding the handling of missing values seems not to be clearly presented. For example, how is the proposed method related to
> > - [Smola et al., Kernel methods for missing variables, 2005]
> > - [Jafrasteh et al., Gaussian processes for missing value imputation, 2023]
> > - [Imani et al, Nested Gaussian process modeling and imputation of high-dimensional incomplete data under uncertainty, 2019]
>
> Thank you for providing these additional references! They are very interesting and insightful. However, we want to point out that **the type of missing values considered in these references differs from our setting**.
>
> **(1) Missing Input Features**
>
> Suppose we are trying to model a latent function with two input dimensions and one output dimension:
>
> (x_1, x_2) --> y
>
> The references which you provided deal with the scenario where, for example, x_1 is missing in some training data examples and x_2 is missing in others. For n examples of d-dimensional training inputs, this corresponds to **missing features in the n x d input data matrix**.
>
> **(2) Missing Output Observations in a Cartesian Product Space**
>
> Our work considers the case of input data which can be expressed as a Cartesian product but is missing output values for some of the input data points.
>
> Suppose we are modeling a latent function
>
> (x_1, x_2) --> y
>
> where x_1 can be a, b, or c, and x_2 can be 1, 2, or 3, such that the whole input space is
>
> | (a, 1) | (a, 2) | (a, 3) |
> |---|---|---|
> | (b, 1) | (b, 2) | (b, 3) |
> | (c, 1) | (c, 2) | (c, 3) |
>
> In our setting, missing values refer to **missing output observations in the structured input space**, for example, missing the y value corresponding to (b, 1) or (c, 3). However, each input observation itself is complete, i.e., we do not consider missing values such as (b, ?) or (?, 3).
>
> In particular, (1) requires imputation of missing values in individual training examples before such data can be used for GPs (or modeling in general). In contrast, (2) does not obstruct the application of GPs in general, it just prevents the use of scalable methods based on Kronecker products. We focus on improving scalability by proposing a method which facilitates the use of Kronecker products in the presence of (2). **Imputing missing values is not the main goal of our contribution.** Therefore, our related work discussion and empirical evaluation focuses on other scalable GP methods, but **we are happy to include a paragraph in the related work section of our camera-ready version which discusses techniques for (1) and clarifies the differences to (2)**.
>
> Smola et al. (2005) also mention "incomplete labels" in a setting with multiple outputs. In the context of GPs, it is possible to treat multiple outputs as an additional (discrete) input dimension:
>
> (x, idx) --> y[idx]
>
> In this case, if we consider the Cartesian product {x} x {idx}, "incomplete labels" would correspond to (2). However, Smola et al. (2005) are proposing a method to integrate these missing labels out, whereas, in our setting, "incomplete labels" would just mean that we have less training data.
>
> ---
>
> > I cannot follow the sentence in pp. 2, " While the above methods scale gracefully to large datasets, they are all are fundamentally limited by performing inference of approximate GP models.". Why are the sparse and variational methods fundamentally limited?
>
> Sparse and variational methods are fundamentally limited because they consider **a small set of inducing points to represent a large number of training data points**. Although these methods are typically designed to minimize the KL divergence to the true posterior ("get as close as possible to the model which uses all training data points"), in general, **they cannot match the true posterior if the number of inducing points is smaller than the number of actual training data points**, where the latter is required to achieve computational benefits. Therefore, "fundamentally limited" refers to the fact that sparse and variational GPs consider inducing point approximations of the true model, where true refers to using all training data points. In contrast, iterative methods consider inference in the true model using numerical techniques to solve the associated linear systems, which are capable of returning the true posterior up to numerical precision.
>
> ---
>
> We believe that we addressed your concerns. If you have follow-up questions, please do not hesitate to ask them. Otherwise, we kindly ask you to consider increasing your score.

---

> > ### Comment · Reviewer_1bD1 · 2025-04-03
> >
> > Thank the authors for the clarifications. I now understand the distinction between the problem addressed in LKGP and those in the references I mentioned. The terms "missing observations/values" are typically used in the literature on recovering missing inputs or targets. As the authors have promised, including a paragraph to clarify these differences would be beneficial. I will raise my score to 3.

---

### Official Review · Reviewer_UJyJ · 2025-03-13

**Overall Recommendation:** 4

**Summary:**

This work proposes an exact latent Kronecker structure for Gaussian Processes, by expressing the covariance matrix of the observed values (not restricted to a grid structure) as the projection of a latent Kronecker product matrix (defined on a grid structure).

Namely, it allows defining a Kronecker product-based matrix structure for the GP covariance, but accommodates observations missing from the assumed, underlying grid.

The combination of such latent Kronecker structure with iterative linear system solvers enables inference over the exact GP model (not an approximation to it), at reduced computational and memory complexity.

The authors demonstrate that the proposed method outperforms state-of-the-art sparse and variational GPs in real-world datasets with many (millions of) examples at reduced complexity.

# After rebuttal
The authors have provided an informative rebuttal to all.

Hence, I argue for acceptance of this submission, and I would encourage the authors to incorporate key aspects of their rebuttal to the camera-ready version.

**Claims And Evidence:**

All their claims are well supported:
- The Latent Kronecker Structure model is sound, and its efficiency is both theoretically (analysis in Proposition 3.1) and empirically (results in Figure 3) demonstrated.

- The computational efficiency and inference quality of the proposed framework are clearly evaluated with experiments in challenging real-word experiments, with many samples.

**Essential References Not Discussed:**

Section 2 of this work presents a very precise and well written overview (plus necessary details) with key references of the Gaussian Process model and the existing different inference frameworks (SparseGP Vs Iterative solvers) needed to situate and understand this work.

**Experimental Designs Or Analyses:**

- The Inverse Dynamics Prediction experiment, with results in Figure 3, is well set-up to fairly compare the matrix multiplication costs of LKGP to baseline iterative methods.

- The Learning Curve Prediction experiment, with results in Figure 4 and Table 1, aims at evaluating performance differences (measured via RMSE, log-likelihood and run-time) under realistic, non-uniform missing values of LKGP and sparse GP alternatives.

- The Climate Data with Missing Values experiment with results in Table 2, showcases the scalability of LKGP to real-life, large datasets with millions of observations: it provides improved performance and reduced computational cost.

**Methods And Evaluation Criteria:**

Absolutely: the experiments are set to evaluate different, yet important criteria of the proposed work (see Experimental Design or Analysis section)

**Other Comments Or Suggestions:**

None

**Other Strengths And Weaknesses:**

Strengths

- The proposed latent Kronecker structure to enable efficient inference in the context of data on grids with missing values is clever and novel.
    - Even if efficient matrix factorization is not possible, the authors figure out how to leverage the projected Kronecker structure for fast matrix multiplication.

- The proposed inference methodology is defined over the exact GP (with assumed covariance structure), rather than over an approximation to the assumed GP prior.

- The proposed method is shown to scale to a large number of training examples, at reduced computational cost.
    - The authors clearly explain and present both the improved asymptotic time and space complexities of matrix
multiplication, as well as the reduced number of kernel evaluations.

Weakness

- The main, yet well discussed and acknowledged weakness of this work is its limitation to GP models defined on a product space, via product kernels.

**Questions For Authors:**

- Can the authors elaborate on why results in Table 1 showcase improved predictive RMSE performance of SVGP, yet not best negative log-likelihood?
    - I understand that LKGP might produce the most sensible uncertainty estimates, yet would appreciate if the authors would elaborate on the reasons/hypothesis for the RMSE gap between LKGP and SVGP

- A key novelty, and critical for the method's efficient use, is the efficient implementation of the latent kernel projections, without explicitly instantiating or multiplying by P, based on zero-padding, slice indexing and lazy evaluations.
    - Can the authors elaborate further on these tricks/techniques, to help unfamiliar readers and for the completeness of the manuscript?

**Relation To Broader Scientific Literature:**

The key contribution of the paper is well defined and presented within the GP inference literature: this work is a highly scalable exact Gaussian process model for product kernels, made possible by leveraging latent Kronecker structure.

**Theoretical Claims:**

The proposed latent Kronecker structure that represents the joint covariance matrix of observed values as a product between projection matrices and a latent Kronecker product is correct to the best of my knowledge.

---

> ### Author Rebuttal · Authors · 2025-03-28
>
> Dear Reviewer UJyJ,
>
> Thank you for your time and effort spent on reviewing our paper. In the following, we provide answers to your specific questions.
>
> ---
>
> > Can the authors elaborate on why results in Table 1 showcase improved predictive RMSE performance of SVGP, yet not best negative log-likelihood?
> > - I understand that LKGP might produce the most sensible uncertainty estimates, yet would appreciate if the authors would elaborate on the reasons/hypothesis for the RMSE gap between LKGP and SVGP
>
> Great question! **Our hypothesis is that LKGP might be overfitting while SVGP might be underfitting**. The setting is prone to overfitting because values are not missing uniformly at random. Instead, missing values are intentionally accumulated at the end of individual learning curves to simulate early stopping, such that **the train (observed) and test (missing) data do not have the same distribution**. Thus, fitting the training data distribution will be prone to overfitting. In particular, LKGP uses the whole training dataset and achieves a tight fit due to the exact GP model, potentially leading to overfitting. SVGP is known to be prone to underfitting, which tends to result in higher observation noise and larger length scales. The latter might actually protect against overfitting in this setting.
>
> In Figure 4, **LKGP achieves a tight fit with high certainty on the observed (solid black) parts of the learning curves, whereas SVGP does not always fit the observed data well** while also estimating a high amount of observation noise and virtually no epistemic uncertainty. The high amount of observation noise can lead to bad NLL despite good mean predictions, while the lack of adequate epistemic uncertainty will penalize the NLL when the mean prediction is inaccurate.
>
> Furthermore, we can compare the performance between train and test sets:
>
> | Average Rank | Train RMSE | Test RMSE | Train NLL | Test NLL |
> |:---:|:---:|:---:|:---:|:---:|
> | LKGP | **1.17** | 2.60 | **1.03** | **1.26** |
> | SVGP | 2.77 | **1.66** | 2.40 | 2.54 |
>
> **LKGP achieves a higher average rank on train RMSE**, which supports the fact that LKGP may be overfitting to the training data distribution, and **SVGP ranks worse in terms of train RMSE**, suggesting that SVGP might be underfitting. The fact that LKGP still achieves the best test NLL is likely due to **superior uncertainty quantification of the exact GP model compared to the variational approximation**, which is a commonly observed phenomenon.
>
> We will add a summary of this discussion to the camera-ready version of our manuscript.
>
> ---
>
> > A key novelty, and critical for the method's efficient use, is the efficient implementation of the latent kernel projections, without explicitly instantiating or multiplying by P, based on zero-padding, slice indexing and lazy evaluations.
> > - Can the authors elaborate further on these tricks/techniques, to help unfamiliar readers and for the completeness of the manuscript?
>
> Thank you for this suggestion! Below, we provide explanations and basic pseudocode examples, which we will also include in the camera-ready version of the manuscript. In practice, we leverage GPyTorch and its underlying linear operator library, which provide more sophisticated implementations.
>
> **Latent Kernel Projections**
>
> In our setting, a projection matrix $P$ is a rectangular matrix which consists of an identity matrix where the rows corresponding to missing values are removed.
>
> For example,
>
> $P = \begin{bmatrix} 1 & 0 & 0 & 0 \\\ 0 & 0 & 1 & 0\end{bmatrix}$,
>
> is a submatrix of a 4 x 4 identity matrix without its 2nd and 4th row.
>
> Multiplying $P$ with a vector removes the 2nd and 4th element.
>
> $P \begin{bmatrix} 1 \\\ 2 \\\ 3 \\\ 4 \end{bmatrix} = \begin{bmatrix} 1 \\\ 3 \end{bmatrix}$
>
> Multipliying $P^T$ with a vector inserts zeros as 2nd and 4th element.
>
> $P^T \begin{bmatrix} 1 \\\ 3 \end{bmatrix} = \begin{bmatrix} 1 \\\ 0 \\\ 3 \\\ 0\end{bmatrix}$
>
> To implement $P$ efficiently, we can store the indices of rows which were not removed from the identity matrix. For the example above, idx = [0, 2].
>
> We can avoid explicit matrix multiplications via
>
> x = [1, 2, 3, 4]
>
> Px = x[idx]
>
> and
>
> x = [1, 3]
>
> PTx = zeros(4)
>
> PTx[idx] = x
>
> **Lazy Kernel Evaluation**
>
> A $n \times n$ kernel matrix $K$ requires $\mathcal{O}(n^2)$ space, which quickly becomes intractable. However, to multiply $K$ with a vector $x$, we do not need to instantiate the full matrix. Instead, given a kernel function which can individually compute each value in $K$, we can iterate over rows of $K$ to perform the calculation using $\mathcal{O}(n)$ space.
>
> for i in range(n): result[i] = k_i @ x
>
> Here, k_i is the i-th row of the kernel matrix, and we rematerialize its values whenever needed to save memory.
>
> ---
>
> We believe that we addressed your questions. If you have follow-up questions, please do not hesitate to ask them. Otherwise, we kindly ask you to consider increasing your score.

---

> > ### Comment · Reviewer_UJyJ · 2025-04-03
> >
> > Dear authors,
> >
> > Thank you very much for your informative responses, which if incorporated to the camera-ready, will significantly improve the manuscript!

---

### Official Review · Reviewer_F5JD · 2025-03-13

**Overall Recommendation:** 4

**Summary:**

In multitask or spatio-temporal Gaussian process regression, scalability issues lead users to adopt a Kronecker, also described as a factored, structure in the covariance between tasks, space, and time. However, this computational gain requires that observations be available for all tasks and inputs, or across all spatial and temporal locations.

In this paper, the authors re-express the product-factored kernel in a setting where only some tasks or locations are observed, formulating it as a down-projection of the complete Kronecker product. They implement this down-projection using zero-padding and slice indexing, ensuring low computational overhead when employing matrix-vector product-based forms of Gaussian process inference.

**Claims And Evidence:**

Yes, the authors claim that their method should have the same predictive performance as standard Gaussian process methods while using less memory and compute time, up until the data points are no longer distributed on a grid, according to the asymptotic break-even point. These claims are substantiated with an extensive experimental comparison.

**Essential References Not Discussed:**

I believe all relevant works were cited.

**Experimental Designs Or Analyses:**

I couldn’t find in the paper whether multiple runs were performed with different selections of missing data for the inverse dynamics experiment.

Additionally, the initialization of hyperparameters and variational parameters is crucial for the predictive performance of both Gaussian process (GP) and sparse GP methods. The fact that GPyTorch’s default initialization was used as-is could negatively affect the results.

I recommend that the authors:

- Clarify the initialization of the variational parameters in SVGP.
- Report the mean and standard deviation over multiple runs of the experiments with different random seeds to ensure more robust results.
- Randomly initialize the Gaussian processes using sensible hyperparameters for each task.

**Methods And Evaluation Criteria:**

Yes, the evaluation criteria make sense. The authors explore different types of datasets (kinematics, training curves, geospatial) and evaluate scenarios with and without uniform gridding.

**Other Comments Or Suggestions:**

Overall, I found the paper easy to read, and the authors have clearly presented and analyzed their approach.

**Other Strengths And Weaknesses:**

None.

**Questions For Authors:**

No additional questions or than the ones raised in the methodology section.

**Relation To Broader Scientific Literature:**

This paper fits well within the broader topic of GP scalability, where the scalability of product-factored kernels has been a common subject of investigation. In particular, this paper contributes to the line of research focusing on GP inference methods that rely solely on matrix multiplications, avoiding Cholesky factorizations and matrix inverses. Within this context, the additional speed-up in matrix multiplication demonstrated by the paper is appreciated.

**Theoretical Claims:**

Yes, the derivation in Appendix A appears correct.

---

> ### Author Rebuttal · Authors · 2025-03-28
>
> Dear Reviewer F5JD,
>
> Thank you for your time and effort spent on reviewing our paper. In the following, we address your suggestions, concerns, and questions.
>
> ---
>
> > I couldn’t find in the paper whether multiple runs were performed with different selections of missing data for the inverse dynamics experiment.
> > - Report the mean and standard deviation over multiple runs of the experiments with different random seeds to ensure more robust results.
>
> Thank you for this suggestion! All three of our experiments on real-world data were performed using different data splits, and **we will include standard errors over multiple runs in the camera-ready version**. In the following, we provide details about our random seed and data split procedures.
>
> **Inverse Dynamics Prediction**
>
> We considered 10 different random splits of the data, as stated in the caption of Figure 3. Each random split considers a different subset of 5000 joint configurations (out of a total of 44484 provided by the full Sarcos dataset) as well as a different pattern of uniformly random missing values.
>
> The relative standard errors (relative to corresponding metric value, averaged across missing ratios) are small, demonstrating stability across replications with different random seeds.
>
> | | Time | Memory | Test RMSE | Test NLL |
> |---|:---:|:---:|:---:|:---:|
> | Iterative GP | 0.85% | 0.00% | 1.27% | 0.33% |
> | LKGP | 0.58% | 0.00% | 1.26% | 0.32% |
>
> **Learning Curve Prediction**
>
> We considered 10 different random seeds, as mentioned in the captions of Table 1 and 3. Each random seed uses the same data but a different pattern of missingness, akin to random train / test splits. In particular, out of n learning curves, we first select missing_ratio x n curves uniformly at random. For each of the selected curves, we choose the early stopping point uniformly at random from [0, t_max - 1], ensuring at least 1 missing value. Thus, for each random seed, the split between complete / incomplete curves is randomized, and the missingness patterns within incomplete curves are also randomized.
>
> Akin to the above, the relative standard errors (averaged across datasets) are small.
>
> | | Test NLL | Test RMSE | Time |
> |---|:---:|:---:|:---:|
> | LKGP | 1.89% | 1.61% | 0.49% |
> | SVGP | 1.24% | 1.29% | 0.25% |
> | VNNGP | 0.71% | 1.09% | 0.26% |
> | CaGP | 3.18% | 1.19% | 0.11% |
>
> **Climate Data**
>
> Due to the large size of the datasets (5M), we only considered 5 different random seeds. The 5M data points themselves are selected once from the NGCD data by drawing 5000 pairs of (latitude, longitude) uniformly at random and taking their corresponding 1000 days of history starting on Jan 1, 2020. For each of the 5 random seeds, the missingness pattern was adapted uniformly at random.
>
> Again, the relative standard errors (averaged across datasets and missing ratios) are small.
>
> | | Test NLL | Test RMSE | Time |
> |---|:---:|:---:|:---:|
> | LKGP | 0.76% | 0.63% | 1.32% |
> | SVGP | 1.11% | 0.29% | 0.09% |
> | VNNGP | 0.13% | 0.27% | 0.40% |
> | CaGP | 1.65% | 0.79% | 0.06% |
>
> ---
>
> > Additionally, the initialization of hyperparameters and variational parameters is crucial for the predictive performance of both Gaussian process (GP) and sparse GP methods. The fact that GPyTorch’s default initialization was used as-is could negatively affect the results.
>
> Thank you for your comment. Regarding kernel hyperparameters and observation noise, we believe that our experimental design is fair, despite using GPyTorch's default initialization, as, within each task, **all GP models use the same kernel paired with the same hyperparameter initialization**. While there might be a better initialization, **it would arguably benefit all GP models**. If certain models require initialization tuning to perform well, this could be considered a downside of that particular model, and similar efforts could be directed towards tuning other models to improve their performance.
>
> Regarding variational parameters / inducing point locations, we initialized them at a uniformly random subset of the training data. This might be suboptimal, and initialization based on, e.g., k-means clustering could potentially lead to better performance. However, given that we consider large datasets of up to 5M data points, **performing clustering or similar algorithms as part of the initialization can become expensive**. For example, building the kNN table for VNNGP required several hours for each of the climate datasets, despite using the highly optimized faiss-gpu implementation (the time was not included in the reported benchmarks).
>
> ---
>
> We believe that we addressed your concerns. If you have follow-up questions, please do not hesitate to ask them. Otherwise, we kindly ask you to consider increasing your score.

---

### Official Review · Reviewer_Zcew · 2025-03-14

**Overall Recommendation:** 3

**Summary:**

The paper improves on the scalability of Kronecker product large scale Gaussian processes (GPs). Specifically,  Kronecker product GPs cannot easily deal with missing observation outputs in the gridded data. The paper proposes a trick based on a projection operation that computes the full Kronecker product kernel matrix and projects it to the sub-matrix  associated with the observed outputs. Somehow this operation still allows for faster matrix-vector multiplications and it can be combined with conjugate gradients solvers to achieve
fast computation for training the GP model or doing inference (e.g., exact sampling from the GP posterior). Three examples mainly in spatio-temporal modeling show that this method can outperform other sparse GP methods in large datasets.

**Claims And Evidence:**

The paper shows convincing evidence that the proposed method can work in low dimensional datasets.
It is unclear if this method is a general purpose GP approximation for high-dimensional inputs.

**Essential References Not Discussed:**

No.

**Experimental Designs Or Analyses:**

The experimental analysis is valid.

**Methods And Evaluation Criteria:**

Yes, the paper uses some standard real-world regression datasets, such as spatio-temporal datasets, which are very
appropriate to demonstrate the proposed method. Also Root Mean Square Error and Test Log Likelihood  are the standard
performance scores for such regression problems.

**Other Comments Or Suggestions:**

The paper is very well written. I didn't notice any typos.

**Other Strengths And Weaknesses:**

Some possible weakness is that the paper does not discuss much the scalability with respect to the dimensionality of the inputs;
see "Questions for Authors" below.

**Questions For Authors:**

The paper is very well written and the proposed method is based on a nice trick using this projection operation illustrated in Figure 1.
Some questions I have are the following:

(i) It is unclear if the method is still applicable in high-dimensional spaces since we know that for Kronecker-product formulations
the number of gridded data (or inducing points) increases exponentially with the number of input dimensions. Does the current method have better scaling with the input dimensionality d?

(ii) Related to the above it is unclear how the authors apply their method to the Sarcos dataset. The Sarcos dataset (see e.g. https://gaussianprocess.org/gpml/data/) has input dimensionality d=21, so did you run using this dimensionality
or much lower? Also these inputs as far as I know are not gridded.  Please state exactly in all datasets which is the dimensionality of the "space" input vector.

(iii) Also it will be useful to add a few more details (e.g., in the Appendix)  about how do you maximize the log marginal likelihood
and particularly how do you deal with the derivatives of the log determinant term.

**Relation To Broader Scientific Literature:**

The paper cites very accurately all the related work in the area.

**Theoretical Claims:**

I checked the derivations and they appear to be correct.

---

> ### Author Rebuttal · Authors · 2025-03-28
>
> Dear Reviewer Zcew,
>
> Thank you for your time and effort spent on reviewing our paper. In the following, we provide answers to your specific questions.
>
> ---
>
> > (i) It is unclear if the method is still applicable in high-dimensional spaces since we know that for Kronecker-product formulations the number of gridded data (or inducing points) increases exponentially with the number of input dimensions. Does the current method have better scaling with the input dimensionality d?
>
> > (ii) Related to the above it is unclear how the authors apply their method to the Sarcos dataset. The Sarcos dataset has input dimensionality d=21, so did you run using this dimensionality or much lower? Also these inputs as far as I know are not gridded. Please state exactly in all datasets which is the dimensionality of the "space" input vector.
>
> Great question! We want to clarify that **our method does not impose a grid over each individual input dimension**. Instead, we assume that the full input data can be expressed as a **Cartesian product of two (or more) factors**, where each factor does not have to be structured and can contain multiple dimensions. Please allow us to explain this in the context of our experiments:
>
> **Inverse Dynamics Prediction**
>
> There are 7 joints with associated position, velocity, acceleration, and torque, and we wish to predict the 7 torques. In our framework, this can be viewed as 22 inputs --> 1 output,
>
> (pos_1, vel_1, acc_1, ..., pos_7, vel_7, acc_7, joint_id) --> torque,
>
> where joint_id ranges from 1 to 7 and specifies which joint's torque the output corresponds to. Here, we partition the input dimensions as
>
> (pos_1, vel_1, acc_1, ..., pos_7, vel_7, acc_7) x (joint_id,),
>
> where the first factor corresponds to an arbitrary configuration of the 7 joints, and joint_id is 1, 2, 3, ..., 7. Therefore, despite joint configurations being 21-dimensional, **the full input data is a Cartesian product with two factors** of size n_configs and 7 respectively, with n_configs = 5000 in our experiments. Crucially, the **joint configurations (pos1, vel1, acc_1, ..., pos_7, vel_7, acc_7) do not need to have any particular structure**.
>
> **Learning Curve Prediction**
>
> We view the problem as
>
> (x_1, ..., x_k, t) --> learning curve value,
>
> and the Cartesian product is
>
> (x_1, ..., x_k) x (t,),
>
> where (x_1, ..., x_k) is a particular hyperparameter configuration and t is the time step of the corresponding training run. Again, **(x_1, ..., x_k) do not need to have any structure and the Cartesian product has two factors**, with n_configs = 2000, k = 7, and t = 1, ..., 52 in our experiments, such that the full input data is of size 2000 x 52 = 104k.
>
> **Climate Data**
>
> We model the problem as
>
> (lat, long, t) --> temperature / precipitation,
>
> and partition the input dimensions as
>
> (lat, long) x (t,),
>
> which again leads to a two-factor Cartesian product, consisting of n_locs = 5000 pairs of non-gridded (lat, long) and t = 1, ..., 1000 days of history per location. The dense input data contains 5000 x 1000 = 5M observed temperature / precipitation values. Notably, **we could also consider latitudes and longitudes on a Cartesian product space**, resulting in an overall three-factor Cartesian product. In general, our method allows an **arbitrary number of factors, where each factor considers a distinct set of input dimensions**.
>
> ---
>
> > (iii) Also it will be useful to add a few more details (e.g., in the Appendix) about how do you maximize the log marginal likelihood and particularly how do you deal with the derivatives of the log determinant term.
>
> Thank you for this suggestion! We have included a section in the appendix to explain marginal likelihood optimization and dealing with corresponding derivatives using iterative methods.
>
> In summary, we use conjugate gradients in combination with fast Kronecker MVMs to solve linear systems, and we estimate the trace term in the derivative of the log det using Hutchinson's trace estimator.
>
> To be more explicit, the derivative of the log marginal likelihood is
>
> $\frac{1}{2} (K^{-1}y)^T \frac{\partial K}{\partial \theta} (K^{-1} y) - \frac{1}{2} tr(K^{-1} \frac{\partial K}{\partial \theta})$,
>
> where $K$ is the kernel matrix, $y$ are the training labels, and $\theta$ are the kernel hyperparameters.
>
> Using Hutchinson's trace estimator, the trace term is approximated as
>
> $\frac{1}{s}\sum_{j=1}^s z_j^T K^{-1} \frac{\partial K}{\partial \theta} z_j$,
>
> where $z_j$ are random probe vectors with $\mathbb{E} [z_j z_j^T] = I$.
>
> To circumvent the explicit inverse of $K$, we instead calculate $K^{-1} y$ and $z_j^T K^{-1}$ by solving $s + 1$ linear systems,
>
> $K [v_y, v_1, ..., v_s ] = [y, z_1, ..., z_s]$,
>
> using conjugate gradients, which only relies on MVMs with $K$ and leverages the Kronecker structure in $K$.
>
> ---
>
> We believe that we addressed your questions and concerns. If you have follow-up questions, please do not hesitate to ask them. Otherwise, we kindly ask you to consider increasing your score.

---

### Decision · Program_Chairs · 2025-05-01

**Decision:**

Accept (poster)

**Comment:**

Thanks to the authors for their interesting and high quality submission.  Reviewers were uniformly positive, and agree that this contribution fills a gap in the scalable Gaussian Process literature.  Reviewers also strongly encourage the authors to incorporate the additional exposition and experiments from the rebuttal period into the final draft of the manuscript.